# Initiation of secondary ice production in clouds

Sylvia C. Sullivan[1], Corinna Hoose[2], Alexei Kiselev[2], Thomas Leisner[2], and Athanasios Nenes[1,3,4,5]

[1]School of Chemical and Biomolecular Engineering, Georgia Institute of Technology, Atlanta, GA 30332, USA
[2]Institute of Meteorology and Climate Research, Karlsruhe Institute of Technology, Karlsruhe, Germany
[3]School of Earth and Atmospheric Sciences, Georgia Institute of Technology, Atlanta, GA 30332, USA
[4]ICE-HT, Foundation for Research and Technology, Hellas, 26504 Patras, Greece
[5]IERSD, National Observatory of Athens, P. Penteli, 15236, Athens, Greece

**Correspondence:** A. Nenes (athanasios.nenes@gatech.edu) and S. Sullivan (scs2229@columbia.edu)

**Abstract.** Disparities between the measured concentrations of ice-nucleating particles (INP) and in-cloud ice crystal number concentrations (ICNC) have led to the hypothesis that mechanisms other than primary nucleation form ice in the atmosphere. Here, we model three of these secondary production mechanisms – rime splintering, frozen droplet shattering, and ice-ice collisional breakup – with a six-hydrometeor-class parcel model. We perform three sets of simulations to understand temporal evolution of ice hydrometeor number ($N_{ice}$), thermodynamic limitations, and the impact of parametric uncertainty when secondary production is active. Output is assessed in terms of the number of primarily nucleated ice crystals that must exist before secondary production initiates ($N_{INP}^{(lim)}$), as well as the ICNC enhancement from secondary production and the timing of a 100-fold enhancement. $N_{ice}$ evolution can be understood in terms of collision-based non-linearity and the 'phasedness' of the process, i.e., whether it involves ice hydrometeors, liquid ones, or both. Ice-ice collisional breakup is the only process for which a meaningful $N_{INP}^{(lim)}$ exists (0.002 L$^{-1}$ up to 0.15 L$^{-1}$). For droplet shattering and rime splintering, a warm enough cloud base temperature and modest updraft are the more important criteria for initiation. The low values of $N_{INP}^{(lim)}$ here suggest that, under appropriate thermodynamic conditions for secondary ice production, perturbations in CCN concentrations are more influential on mixed-phase partitioning than those in INP concentrations.

## 1 Background

Number concentrations of ice-nucleating particles ($N_{INP}$) in the atmosphere span orders of magnitude from a few per cubic meter up to 100s per liter (e.g., DeMott et al., 2010). At temperatures greater than about -15°C, these concentrations remain low: only one particle in every $10^3$ or $10^4$ will nucleate an ice crystal (Rogers et al., 1998; Chubb et al., 2013; DeMott et al., 2015). However, even when INP concentrations are low at warm subzero temperatures, in-cloud ice crystal number concentrations (ICNC) can be orders of magnitude higher (e.g., Hallett and Mossop, 1974; Heymsfield and Willis, 2014; Lasher-Trapp et al., 2016; Taylor et al., 2016; Ladino et al., 2017), particularly in tropical maritime clouds (Koenig, 1963, 1965; Hobbs and Rangno, 1990).

This discrepancy may be explained in some cases by shattering upon cloud probe tips (Field et al., 2003; Heymsfield, 2007; McFarquhar et al., 2007), but even as instrumentation and algorithms have been developed to minimize these artifacts (Korolev

et al., 2013; Korolev and Field, 2015), the disparity has remained, supporting several hypothesized secondary ice production processes. Hallett and Mossop (1974) proposed rime splintering in which ice hydrometeors collide with and freeze supercooled droplets to form rime, which then splinters off as the hydrometeor continues to fall. Droplets in cases of rime splintering tend to be both less than 13 $\mu$m and greater than 25 $\mu$m in diameter, and temperatures fall between -3 and -8°C (Mossop, 1978; Heymsfield and Mossop, 1984; Mossop, 1985); however, ICNC enhancement, i.e., an increase in ICNC beyond that generated by primary nucleation, exists even outside of these conditions.

Another hypothesized mechanism is the shattering of droplets with a diameter of 50 to 100s of $\mu$m upon freezing (Mason and Maybank, 1960; Cannon et al., 1974; Korolev et al., 2004; Fridlind et al., 2007; Rangno, 2008; Leisner et al., 2014; Lawson et al., 2015; Wildeman et al., 2017). At sufficiently cold temperatures, latent heat release leads to the formation of a liquid core-ice shell structure that eventually shatters upon internal pressure build-up. A third mechanism, independent of the liquid phase, is breakup upon mechanical collision of ice hydrometeors. Vardiman (1978) calculated the fragment number generated during ice-ice collisional breakup from a change in momentum, and Takahashi et al. (1995) later conducted experiments with a rotating ice sphere in a cloud chamber to estimate the number of ice crystals ejected versus temperature. Yano and Phillips (2011), and more recently Yano et al. (2015), have identified 'explosive regimes' defined by non-dimensional parameters, where ice-ice collisional breakup may enhance ICNC by as much as $10^4$.

Laboratory and in-situ data of these processes are difficult to obtain, and their fragment generation functions and temperature dependence remain uncertain (Field et al., 2017). Given these uncertainties, implementation of secondary ice production parameterizations in large-scale models is still premature. Instead, small-scale , more controllable modeling provides a good tool to estimate variability of secondarily-produced ICNC with these parameters, as well as the minimum number of INP needed to initiate secondary production. We call this latter variable $N_{\mathrm{INP}}^{(lim)}$ hereafter.

Some previous studies have estimated $N_{\mathrm{INP}}^{(lim)}$ on the basis of in-situ data. For example in a study of ice initiation in cumulus, Beard (1992) found that a nucleated ICNC of 0.001 L$^{-1}$ could trigger raindrop freezing around -5°C . More recently, Crawford et al. (2012), with Aerosol Properties, PRocesses And InFluenceS (APPRAISE) campaign data, and Huang et al. (2017), with Ice and Precipitation Initiation in Cumulus (ICEPIC) campaign data, identified a primarily nucleated ICNC of 0.01 L$^{-1}$ as sufficient to initiate rime splintering. Connolly et al. (2006) found that the rime splintering tendency increased with increasing primarily-nucleated ICNC, but this result was based upon adjusting the primary nucleation rate rather than the absolute $N_{\mathrm{INP}}$. Clark et al. (2005) also adjusted the primary nucleation rate relative to the rime splintering one, but gave no approximate $N_{\mathrm{INP}}^{(lim)}$ values or thermodynamic constraints. These studies have also considered only rime splintering, despite evidence that multiple processes occur simultaneously (Rangno and Hobbs, 2001). We provide more comprehensive estimates of $N_{\mathrm{INP}}^{(lim)}$ here for three secondary production processes over a range of thermodynamic conditions and fragment numbers.

## 2 Parcel model

To estimate ICNC enhancement and $N_{\mathrm{INP}}^{(lim)}$, we run a parcel model with six hydrometeor classes for small ice crystals and droplets, small and large graupel, and medium and large droplets (Sullivan et al., 2017). The number in these classes is denoted

$N_i$, $N_d$, $N_g$, $N_G$, $N_r$, and $N_R$ respectively. The hydrometeors in each class are assumed monodisperse, but their sizes are tracked over time as a function of temperature and superaturation. $N_{ice}$ is used to denote the summation of the number in the three ice hydrometeor classes. The bin microphysics consists of primary nucleation and secondary production by ice-ice collisional breakup, rime splintering, and frozen droplet shattering. These processes are included in an ice generation function

with units of m$^{-3}$ s$^{-1}$:

$$G_{ice} = \left.\frac{dN_i}{dt}\right|_{NUC} \qquad + \left.\frac{dN_i}{dt}\right|_{BR} \qquad\qquad + \left.\frac{dN_i}{dt}\right|_{RS} \qquad\qquad\qquad\qquad\qquad + \left.\frac{dN_i}{dt}\right|_{DS} \tag{1}$$

$$= c_0 f_{imm} \qquad + \eta_{BR} K_{BR} \aleph_{BR} N_g N_G \qquad + \eta_{RS} \aleph_{RS} \left[ K_{RS,g} N_g + K_{RS,G} N_G \right] N_R \qquad + \eta_{DS} \aleph_{DS} N_R \tag{2}$$

$NUC$ stands for nucleation, $BR$ for ice-ice collisional breakup, $RS$ for rime splintering, and $DS$ for droplet shattering. $c_0$ is the primary nucleation rate derived from the temperature dependence of the immersion INP concentration given in DeMott

et al. (2010); $\eta_X$ is the weighting for process $X$, either 100% when the process is active or 0% when it is inactive; $K_X$ is a gravitational collection kernel for process $X$; and $\aleph_X$ is the fragment number generated by process $X$. More specifically, the nucleation rate is calculated as the product of updraft velocity, an assumed lapse rate of 6 K km$^{-1}$, and the temperature derivative of the INP concentration: $u_z \Gamma d/dT [a_1 \exp(a_2(T - a_3))]$. The factor $f_{imm}$ indicates the fraction of these INP that nucleate immediately in cloud droplets. This formulation requires no explicit treatment of aerosol.

Expressions for $\aleph_X$ are given in Table 1: $\aleph_{RS}$ is taken from the laboratory experiments of Hallett and Mossop (1974), and $\aleph_{BR}$ is based upon those of Takahashi et al. (1995). $\eta_{RS}$ is set to 1% outside an optimal temperature zone of -3 to -8°C to allow for cases in which local temperature gradients may still permit rime splintering at the hydrometeor surface. $\aleph_{DS}$ contains a product of droplet freezing and shattering probabilities, $p_{fr}$ and $p_{sh}$, and either polynomial (as in Lawson et al. (2015)) or sigmoidal dependence on large droplet size. $p_{fr}$ synthesizes the INP concentration from DeMott et al. (2010) with the particle

accumulation simulations of Paukert et al. (2017). The remaining INP that do not nucleate immediately as given by $f_{imm}$ are found in cloud droplets that undergo about 100 collisions before they form a "shatter-able" drop. $p_{sh}$ is non-zero for drops of radii greater than 50 $\mu$m and has Gaussian temperature dependence centered at 258 K on the basis of droplet levitation experiments. The droplet shattering tendency is later modified to represent a collisional process with a product of large droplet and ice crystal numbers and then denoted $DScoll$.

For the liquid phase, a droplet generation function consists simply of droplet activation, calculated from a Twomey power-law formulation. Thie formulation is supersaturation-dependent and, again, requires no explicit aerosol. The number balance in each class is then the generation function at the current time as a source and the generation function at a time delay as the sink, along with aggregation and coalescence losses. For example, the number in the ice crystal class is given by

$$\frac{dN_i}{dt} = G_{ice}(t) - G_{ice}(t - \tau_i) - \eta_{agg} K_{agg} N_i N_g. \tag{3}$$

The time delays, $\tau_X$, quantify how long depositional, riming, or condensational growth to the next hydrometeor class will take and are solved for approximately from growth equations. Newly produced ice crystals are assumed to be spherical with bulk

ice density, while graupel is assumed to be spheroidal with a deposition density and non-unit capacitance as in Chen and Lamb (1994). The coalescence efficiency is assumed to be unity between small and medium droplets (Klett and Davis, 1973). A basic representation of large droplet coalescence is employed at temperatures above 273 K, given the importance of droplet size distribution broadening to droplet shattering (Lawson et al., 2017): $N_d$ is reduced by 5% every minute due to coalescence, and the mass is redistributed among the remaining large droplets.

**Table 1.** Default parameter values from simulations and their sources. $\mathcal{N}(\mu,\sigma)$ indicates a normal distribution with mean $\mu$ and standard deviation $\sigma$.

| Parameter | Value | Source |
|---|---|---|
| ***Fragment number*** | | |
| $\aleph_{RS} = F_{RS}\rho_w \frac{\pi}{6}(2r_R)^3$ | $F_{RS}$ = 3 x $10^8$ (kg rime)$^{-1}$ | Hallett and Mossop (1974) |
| $\aleph_{BR} = F_{BR}(T - T_{min})^{1.2}\mathrm{e}^{-(T-T_{min})/5}$ | $F_{BR}$ = 280 | Takahashi et al. (1995) |
| | $T_{min}$ = 252 K | |
| $\aleph_{DS} = F_{DS}(2r_R)^4 p_{fr}(T, r_R, N_d)p_{sh}(T, r_R)$ | $p_{sh} = 0.2\mathcal{N}(258\ \mathrm{K}, 10\ \mathrm{K})$ | Based upon droplet |
| | for $r_R > 50\ \mu$m | levitation experiments |
| | $p_{fr} = 100\,(1 - f_{imm})\big[a_1 \exp(a_2(T - a_3))\big]/N_d$ | |
| | with $f_{imm}$ = 90% | Based upon Paukert et al. (2017) |
| $\aleph_{DS}^{(coll)} = F_{DS}(2r_R)^4 p_{sh}(T)$ | $p_{sh}$ as above | |
| | $F_{DS}$ = 2.5 x $10^{-11}$ | Lawson et al. (2015) |
| | (drop diam [$\mu$m])$^{-4}$ | |
| $\aleph_{DS}^{(sig)} = \dfrac{\alpha\,p_{fr}(t, T, r_R)p_{sh}(T)}{1 + \exp[-\beta(2r_R - \gamma)]}$ | $\alpha$ = 10; $\beta$ = -0.016 | Based upon droplet |
| | $\gamma$ = 500 | levitation experiments |
| ***Initial conditions*** | | |
| $N_{X0}$ | 0 cm$^{-3}$ | |
| $P_0, s_{w,0}$ | 680 hPa, $10^{-6}$% | |
| $r_{d0}, r_{r0}, r_{R0}$ | 1, 12, 25 $\mu$m | Mossop (1978, 1985) |
| $r_{i0}, a_{g0}, a_{G0}$ | 5, 50, 200 $\mu$m | Zhang et al. (2014) |
| | | Reinking (1975) |
| ***Time scales*** | | |
| $\tau_d, \tau_r, \tau_R$ | 5, 15, 25 min | Approximate solution |
| $\tau_i, \tau_g, \tau_G$ | 7.5, 20, 17.5 min | of growth equations |
| ***Droplet spectrum*** | | |
| $k_{CCN}, N_{CCN}$ | 0.308, 100 cm$^{-3}$ | (Hegg et al., 1992) |
| ***Updraft*** $\quad u_z$ | 2 m s$^{-1}$ | (Korolev and Field, 2007) |
| ***Time step*** $\quad \Delta t$ | 3 sec | |

The six hydrometeor number tendencies are solved with an explicit Runge-Kutta (2,3) pair for delay differential equations (Bogacki and Shampine, 1989) and coupled to moist thermodynamic equations for pressure, temperature, supersaturation, mixing ratios, and hydrometeor sizes. This second set of equations is solved with a Rosenbrock formula of order 2 (Rosenbrock,

1963). The model microphysics is shown schematically in Figure S1, and parameter values and sources are given in Table 1. Model assumptions, thermodynamic tendencies and correlations, and collection kernels are more thoroughly described in Sullivan et al. (2017) and their effects more thoroughly discussed below in Section 4.

## 3   Simulations

The three rows of Table 2 show three sets of simulations with the parcel model. First we investigate the evolution of the total ice hydrometeor number, $N_{ice}$, i.e., the summation of $N_i$, $N_g$, and $N_G$, in default simulations with fixed fragment numbers and thermodynamic conditions. Simulation acronyms include $BR$ for ice-ice collisional breakup, $DS$ for droplet shattering, or $RS$ for rime splintering are active. These runs address how the value of $N_{INP}^{(lim)}$ and enhancement magnitude or timing vary when different processes are active. We quantify enhancement from secondary production as the ratio of the total ICNC to the number generated by primary nucleation when the simulation ends, i.e., when the parcel becomes water subsaturated or reaches a temperature of 237 K above which no homogeneous nucleation occurs: $N_{ice}(t_{end})/N_{INP}(t_{end})$. An enhancement of 10 can be understood as *at least* a 10-fold increase in ICNC due to secondary production, as an aggregation sink is also active in the simulations. In the absence of secondary production, ICNC enhancement does not exceed one.

The second set of simulations considers the effect of updraft velocity and initial temperature in the parcel; this set is denoted 'th' for thermodynamics. The updraft is varied from 0.1 up to 5 m s$^{-1}$ to simulate both stratiform and convective conditions, while the initial parcel temperature is adjusted from a quite warm cloud base temperature of 295 K down to the temperature at the droplet shattering probability peak of 256 K. Then parameter perturbations are performed in a final set, denoted 'pp'. In particular, we vary the leading coefficient of the fragment number generated per collision and per kilogram of rime, $F_{BR}$ and $F_{RS}$ respectively; the minimum temperature for which ice-ice collisional breakup occurs, $T_{min}$; the functional form of the fragment number generated per shattering droplet, $F_{DS}(\beta, \gamma)$; and the maximum of the temperature-dependent droplet shattering probability distribution, $p_{sh}^{(max)}$. The effect of these parameters on the generated fragment numbers is shown in Figure S2, and the alternate sigmoid functions for $\aleph_{DS}$ are shown in Figure S3.

### 3.1   Hydrometeor number evolution

The temporal evolution of $N_{ice}$ in the default simulations is shown in Figure 1. Each simulation is done for a range of total INP number within the parcel, $N_{INP}^{(tot)}$. A base run with only nucleation shows the $N_{INP}^{(tot)}$ thresholds in panel **(d)**. The structure in the number evolution can be understood by considering whether the process is collisional and its 'phasedness', i.e., whether it involves hydrometeors in the liquid or ice phase or both. The ice mass mixing ratio and ice crystal radius evolution are also shown in Figures S5 and S6, but analysis focuses on $N_{ice}$ below.

When the process involves a product of hydrometeor numbers, as for breakup and rime splintering, the $N_{ice}$ evolution is non-linear. Independent of $N_{INP}^{(tot)}$, $N_{ice}$ grows steadily throughout the simulation for these collisional secondary production processes. Even as graupel or large droplets are consumed, those hydrometeors still in the parcel continue to grow by deposition or condensation respectively. This ongoing hydrometeor growth increases the secondary production tendencies via their

**Table 2.** All simulations with parameters adjusted from the default values in Table 1. A control run with no secondary production, i.e., $\eta_{DS} = \eta_{BR} = \eta_{RS} = 0\%$, is denoted INP in Figure 1. Simulations run with combinations (BRDS, BRRS, and DSRS) or all (ALL and ALLth) of the processes are shown in the Supplement and detailed in Table S1.

| *Run BR* | *Run RS* | *Run DS* |
| --- | --- | --- |
| | | *(Run DScoll)* |
| Ice-ice collisional breakup | Rime splintering | Droplet shattering |
| | | (Collisional droplet shattering) |
| $\eta_{DS} = \eta_{RS} = 0\%$ | $\eta_{BR} = \eta_{DS} = 0\%$ | $\eta_{BR} = \eta_{RS} = 0\%$ |

| *Run BRth* | *Run RSth* | *Run DSth* |
| --- | --- | --- |
| Thermodynamic variations | Thermodynamic variations | Thermodynamic variations |
| for ice-ice collisional breakup | for droplet shattering | for rime splintering |
| $u_z = \{\ 0.1,\ \ 0.5,\ \ 1,\ \ 1.5,\ \ 2,\ \ 2.5,\ \ 3,\ \ 3.5,\ \ 4\ \mathrm{m\ s^{-1}}\}$ | | |
| $T_0 = \{\ 256, 258, 260, 262, 264, 268, 270, 272\ \mathrm{K}\}$ | | $T_0 = \{\ 272, 275, 280, 285, 288, 290, ...$ |
| | | $293, 295, 298\ \mathrm{K}\ \}$ |

| *Run BRpp* | *Run RSpp* | *Run DSpp* |
| --- | --- | --- |
| Parameter perturbations | Parameter perturbations | Parameter perturbations |
| for ice-ice collisional breakup | for rime splintering | for droplet shattering |
| $F_{BR} = \{0, 90, 140, 200, 280\}$ | $F_{RS} = \{9, 15, 30, 45, 80\}$ | $F_{DS} = \{25, 75\}\mathrm{x}\ 10^{-12}(2\,r_D)^{-4\ \mathrm{or}\ -3}$ |
| | x $10^7$ (kg rime)$^{-1}$ | $(\beta, \gamma) = \{\ (-0.016, 500), (-0.015, 400)\}$ |
| $T_{min} = \{246, 249, 252, ...$ | | $p_{sh}^{(max)} = \{1, 5, 10, 20, 30\%\}$ |
| $255, 258\ \mathrm{K}\}$ | | |

collection kernels, and this link itself is non-linear because both hydrometeor terminal velocity and collisional cross section increase with growth. This idea is shown qualitatively in the red and blue traces of Figure 8a.

When the process involves a single hydrometeor number, as for droplet shattering here, the $N_{\text{ice}}$ evolution is almost linear and increases suddenly. This threshold behavior occurs when the temperature becomes cold enough for non-negligible shattering and freezing probabilities. Although these factors control the initiation, the fragment and large droplet numbers, $\aleph_{\text{DS}}$ and $N_R$, control the enhancement magnitude from the droplet shattering. As a result, the traces in Figure 1b do not exhibit direct dependence on $N_{\text{INP}}^{(tot)}$: all $DS$ simulations reach an $N_{\text{ice}}^{(max)}$ of 29 L$^{-1}$ in about 46 minutes. But the implicit dependence on a nonzero $N_{\text{INP}}^{(tot)}$ should be pointed out: in the absence of INP, $p_{fr}$ is always zero and no droplet shattering occurs. As soon as $p_{fr}$ becomes nonzero, the other terms in the droplet shattering tendency are large enough to produce an ICNC enhancement. Below in Section 3.1.1, we discuss cloud base temperature dependence and a collisional mechanism ($DScoll$). The $DS$ and $DScoll$ trends are also shown qualitatively in the green traces of Figure 8a.

Because breakup and rime splintering involve the ice phase, increasing $N_{\text{INP}}^{(tot)}$ boosts their rates of generation and yields large enhancement sooner. For ice-ice collisional breakup, a parcel with 0.0129 L$^{-1}$ INP reaches 10 L$^{-1}$ $N_{\text{ice}}$ in 23 minutes, while that with 0.167 L$^{-1}$ INP reaches the same value in 17 minutes. For rime splintering, the same increase in INP shifts the time to reach 10 L$^{-1}$ $N_{\text{ice}}$ from 30 minutes back to 25. While these differences in enhancement timing sound small, they can help infer which secondary production processes are active from in-situ $N_{\text{INP}}$ and ICNC data. For example, ICNC on the order of hundreds per liter can form within 10 to 15 minutes (Hobbs and Rangno, 1990; Rangno and Hobbs, 1991, 1994). This timing is too rapid to be explained by rime splintering alone (Mason, 1996), in agreement with our $RS$ simulation. Simulations with ice-ice collisional breakup and rime splintering in combination, on the other hand, *are* sufficiently rapid (Fig. S4b).

Higher $N_{\text{INP}}^{(tot)}$ only increases the ice generation rates from ice-ice collisional breakup and rime splintering up to a certain point however. Beyond an $N_{\text{INP}}^{(tot)}$ of about 0.599 L$^{-1}$, additional INP do not increase $N_{\text{ice}}^{(max)}$. The parcel is in a supersaturation-limited regime, for which it becomes subsaturated before the effect of additional primary nucleation can be felt by secondary production.

Finally non-linearity and hydrometeor phases involved determine enhancement magnitude. The ice-ice collisional breakup tendency is proportional to the product of two ice hydrometeor numbers, $N_g$ and $N_G$, so the impact of varying $N_{\text{INP}}^{(tot)}$ is most pronounced for the $BR$ simulations. Increasing $N_{\text{INP}}^{(tot)}$ by two orders of magnitude (0.001 to 0.167 L$^{-1}$) increases $N_{\text{ice}}^{(max)}$ by four order of magnitude (0.0023 to 37.6 L$^{-1}$). The rime splintering and droplet shattering tendencies are proportional to $N_R$ which is around $10^6$ times as large as $N_g$ or $N_G$, so the impact of $N_{\text{INP}}^{(tot)}$ for these processes is diluted. For the purely liquid-phase droplet shattering, the two-order-of-magnitude increase in $N_{\text{INP}}^{(tot)}$ has no significant impact on $N_{\text{ice}}^{(max)}$. For rime splintering, it actually translates to a two-fold decrease in $N_{\text{ice}}^{(max)}$ (30.58 to 16.67 L$^{-1}$). This decrease is the result of an increasing denominator in the $N_{\text{ice}}^{(max)}/N_{\text{INP}}(t_{end})$ expression (see also the RS panels of Figures 3 and 4 below). The rime splintering tendency is strong enough that it always generates additional ice crystals, so increasing $N_{\text{INP}}^{(tot)}$ actually decreases enhancement. The total INP number does, however, affect which rimers contribute to enhancement: when $N_{\text{INP}}^{(tot)}$ exceeds 0.167 L$^{-1}$, only rime splintering of small graupel can occur before subsaturation of the parcel.

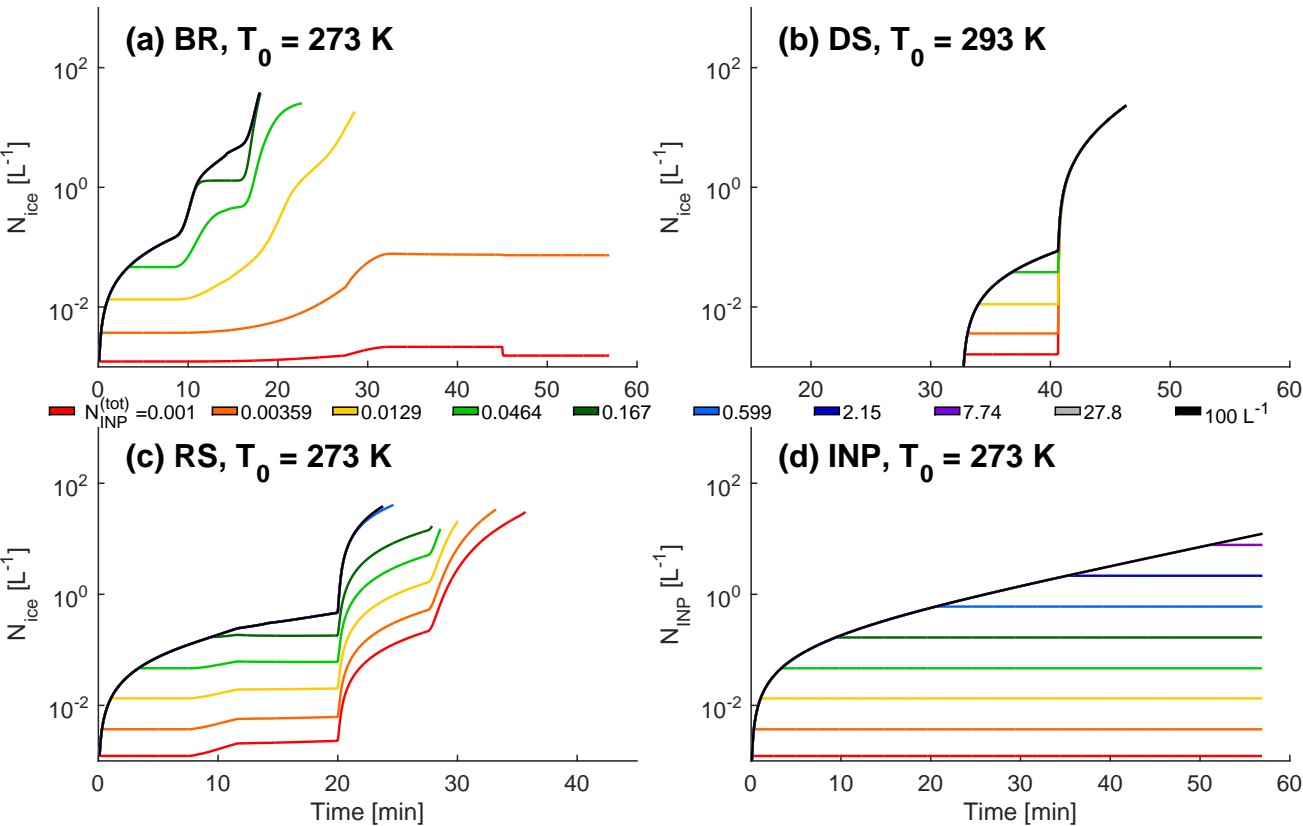

**Figure 1.** Evolution of the total ice hydrometeor (summation of ice crystal, small and large graupel numbers) number for default simulations with a range of $N_{INP}^{(tot)}$ from 0.001 L$^{-1}$ up to 100 L$^{-1}$: **(a)** ice-ice collisional breakup only, **(b)** droplet shattering only, **(c)** rime splintering only, and **(d)** a control run when only primary nucleation is active. These default simulations are run for $u_z$ of 2 m s$^{-1}$ and $T_0$ values given in each panel.

### 3.1.1 Droplet shattering formulation

For droplet shattering, we additionally investigate the impact of $T_0$ and the underlying physical mechanism. First, recent experimental evidence indicates the importance of warm cloud base temperature and the warm rain process to any subsequent droplet shattering (Lawson et al., 2015; Taylor et al., 2016; Lawson et al., 2017). Although the model with its six monodisperse hydrometeor classes cannot fully represent droplet size distribution broadening, we implement simplified large droplet coalescence whereby the large droplet number is reduced by 5% every minute above the freezing level. Liquid mass is conserved by redistribution among the remaining drops. Without this process, the condensational growth from a diameter of 50 $\mu$m up to about 100 $\mu$m is very slow, and an appropriate dependence on $T_0$ is not reproduced.

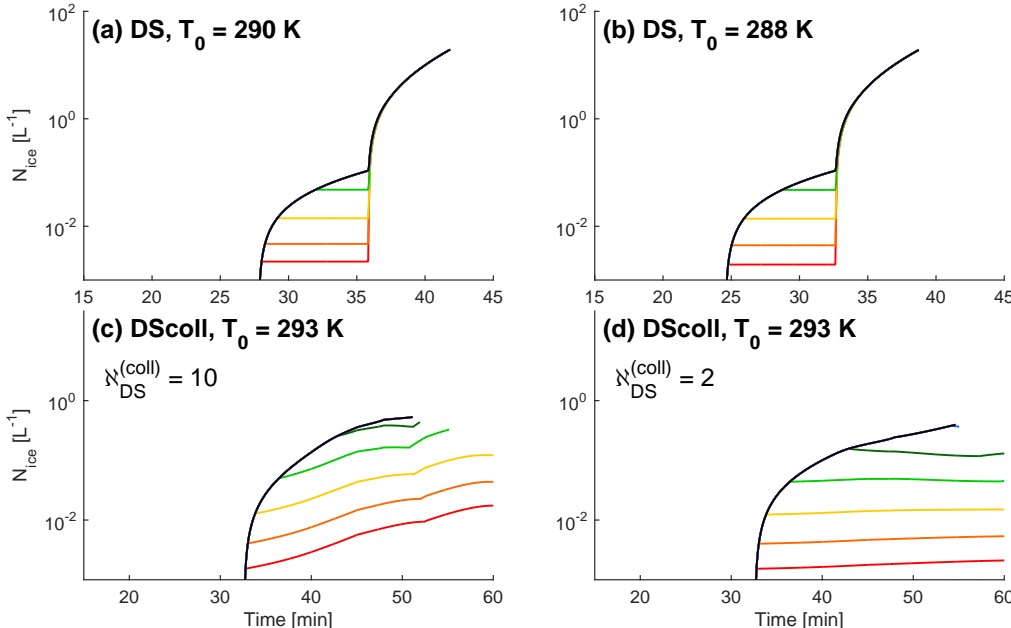

**Figure 2.** Evolution of $N_{\text{ice}}$ at different cloud base temperatures, 290 K in panel **(a)** and 288 K in panel **(b)**, and a collisional formulation with **(c)** 10 fragments generated ($\aleph_{DS}^{(coll)} = 10$) and **(d)** 2 fragments generated per collision ($\aleph_{DS}^{(coll)} = 2$). Otherwise, $p_{sh,max} = 20\%$, $u_z = 2$ m s$^{-1}$ as in Figure 1. Coloring indicates the $N_{\text{INP}}^{(tot)}$ value as in Figure 1.

Figures 2a and b show the impact of lowering $T_0$ to 290 or 288 K from the 293 K level shown in Figure 1. The colder this cloud base temperature becomes, the sooner the temperature threshold for non-negligible $p_{fr}$ and $p_{sh}$ is reached. But as $T_0$ drops, the resultant $N_{\text{ice}}^{(max)}$ also drops. At 293 K, $N_{\text{ice}}$ of 29.2 L$^{-1}$ is produced in 46.3 minutes; at 290 K, 19.4 L$^{-1}$ in 41.8 minutes; and at 288 K, 15.1 L$^{-1}$ in 38.7 minutes. Once $T_0$ drops to 285 K or below at this updraft, droplet shattering is

no longer effective because the large droplets do not have sufficient time to grow, either by coalescence or condensation, to a size at which they can shatter. There is also an upper bound to warmer $T_0$. At 298 K, $N_{\text{ice}}^{(max)}$ increases to 86.1 L$^{-1}$ after 53 minutes, but beyond that, the droplets begin to sediment before the parcel reaches sufficiently cold temperatures for their shattering. These $u_z - T_0$ dependencies are shown qualitatively in Figures 8b and Figure 9 and described in greater in Section 3.2 below.

Then, the exact mechanism underlying this droplet shattering remains uncertain, and it has been hypothesized that it initiates via collision between a large droplet and a small ice crystal. In this case, the droplet shattering tendency would be proportional to both $N_R$ and $N_i$, rather than just $N_R$, in the final term of Equation 2:

$$\left.\frac{dN_i}{dt}\right|_{DS} = \eta_{DS} \aleph_{DS}^{(coll)} K_{DS} N_R N_i \tag{4}$$

The fragment number from Lawson et al. (2015) ($F_{DS} D_R^4$) and $p_{sh}$ are retained as in the DS simulation, but $p_{fr}$ is removed

with the understanding that collision with the ice surface initiates freezing.

In Figures 2c and d, the threshold behavior of the enhancement in the $DS$ simulation is replaced by a steady increase similar to that from the $RS$ or $BR$ simulation in Figure 1. This formulation yields a smaller $N_{\text{ice}}^{(max)}$ than $DS$, up to 0.39 L$^{-1}$ when $\aleph_{DS}^{(coll)} = 2$ or 0.53 L$^{-1}$ when $\aleph_{DS}^{(coll)} = 10$. Enhancement timing is also slowed with $N_{\text{ice}}^{(max)}$ reached after 54 minutes for $\aleph_{DS}^{(coll)} = 2$ and 51 minutes for $\aleph_{DS}^{(coll)} = 10$. The simultaneous consumption and generation of crystals during collisions now means that $dN_i/dt \propto N_i$, and the process will never generate the superexponential increases as from $BR$ in Figure 1a. The dual source-sink of $N_i$ also means that $N_{\text{INP}}^{(tot)}$ has a large effect on enhancement magnitude. From an $N_{\text{INP}}^{(tot)}$ of 0.001 L$^{-1}$ up to 0.6 L$^{-1}$, there is a two order-of-magnitude-difference in the ultimate $N_{\text{ice}}$. However, given the slower $N_{\text{ice}}$ generation rate of this collisional process, initiation of non-collisional droplet shattering by immersed INP is likely to be the more influential process.

## 3.2 Varying thermodynamics

Secondary enhancement from the simulations with varying thermodynamics are shown in Figures 3 and 4. Runs are performed for a range of updraft velocities and initial temperatures given in Table 2, but we focus on the extremes.

The top panels of Figure 3 show enhancement for stratiform conditions, i.e., $u_z$ of 0.5 m s$^{-1}$, and a range of cloud base temperatures $T_0$. Clear $N_{\text{INP}}^{(lim)}$ values for ice-ice collisional breakup can be seen in panel (a). As $T_0$ decreases from 272 to 270 to 268 K, $N_{\text{INP}}^{(lim)}$ drops from 32.8 to 21.5 to 2.1 m$^{-3}$. At 266 K, $N_{\text{INP}}^{(lim)}$ increases again, reaching an $\mathcal{O}(10^2)$ enhancement only for an INP concentration of 0.143 L$^{-1}$. Larger ICNCs occur only at these warmer $T_0$ because the parcel remains in the mixed-phase temperature range long enough that large graupel can form by riming (see also Figure 8b). For rime splintering, there is no $N_{\text{INP}}^{(lim)}$ value greater than 1 m$^{-3}$: the enhancement is largest at the lowest value of $N_{\text{INP}}^{(tot)}$ in Figure 3 and decreases with higher values of $N_{\text{INP}}^{(tot)}$.

Then when $u_z$ is increased to 4 m s$^{-1}$ in panel (d), rime splintering occurs over an expanded range of $T_0$ but with a reduced enhancement magnitude. As the parcel moves faster, it is more likely to pass through the optimal 'RS temperature zone' of 267 to 269 K or obtain higher $p_{sh}$ or $p_{fr}$, but it also spends less time in these optimal zones. This idea is also true for the $DSth$ runs shown in panel (d): a faster-moving parcel must initiate at a warmer temperature for sufficiently large droplets to form before the freezing level. No enhancement occurs from ice-ice collisional breakup at these faster updrafts because there is always insufficient time for graupel to form. The general favorability of modest updrafts is shown in Figure 8b.

If instead, we fix $T_0$ and look at a range of $u_z$ as in Figure 4, ice-ice collisional breakup remains the only process with a defined $N_{\text{INP}}^{(lim)}$. This threshold value decreases from 32.8 m$^{-3}$ at 0.5 m s$^{-1}$ down to 1.52 m$^{-3}$ at 1.5 m s$^{-1}$. At 2.5 m s$^{-1}$, it increases back up to 50 m$^{-3}$, and at the fastest updraft velocities, no enhancement from ice-ice collisional breakup occurs again because graupel does not form. In this case, not only is the parcel too short-lived for graupel formation; diffusional growth is also slowed significantly at such low temperatures.

Although there is no meaningful $N_{\text{INP}}^{(lim)}$ for droplet shattering or rime splintering, $N_{\text{INP}}$ still affects enhancement from these processes. In fact, increasing $N_{\text{INP}}^{(tot)}$ generally decreases enhancement for all $u_z - T_0$ conditions. This can be understood in terms of a sort of **INP efficiency**: the highest ICNC per INP is produced when $N_{\text{INP}}^{(tot)}$ is lowest. Mathematically, increasing $N_{\text{INP}}^{(tot)}$ increases the denominator of the enhancement ratio without a corresponding increase in the numerator. Physically, a higher $N_{\text{INP}}^{(tot)}$ depletes supersaturation more rapidly, as many small ice crystals grow by deposition, or it may keep the

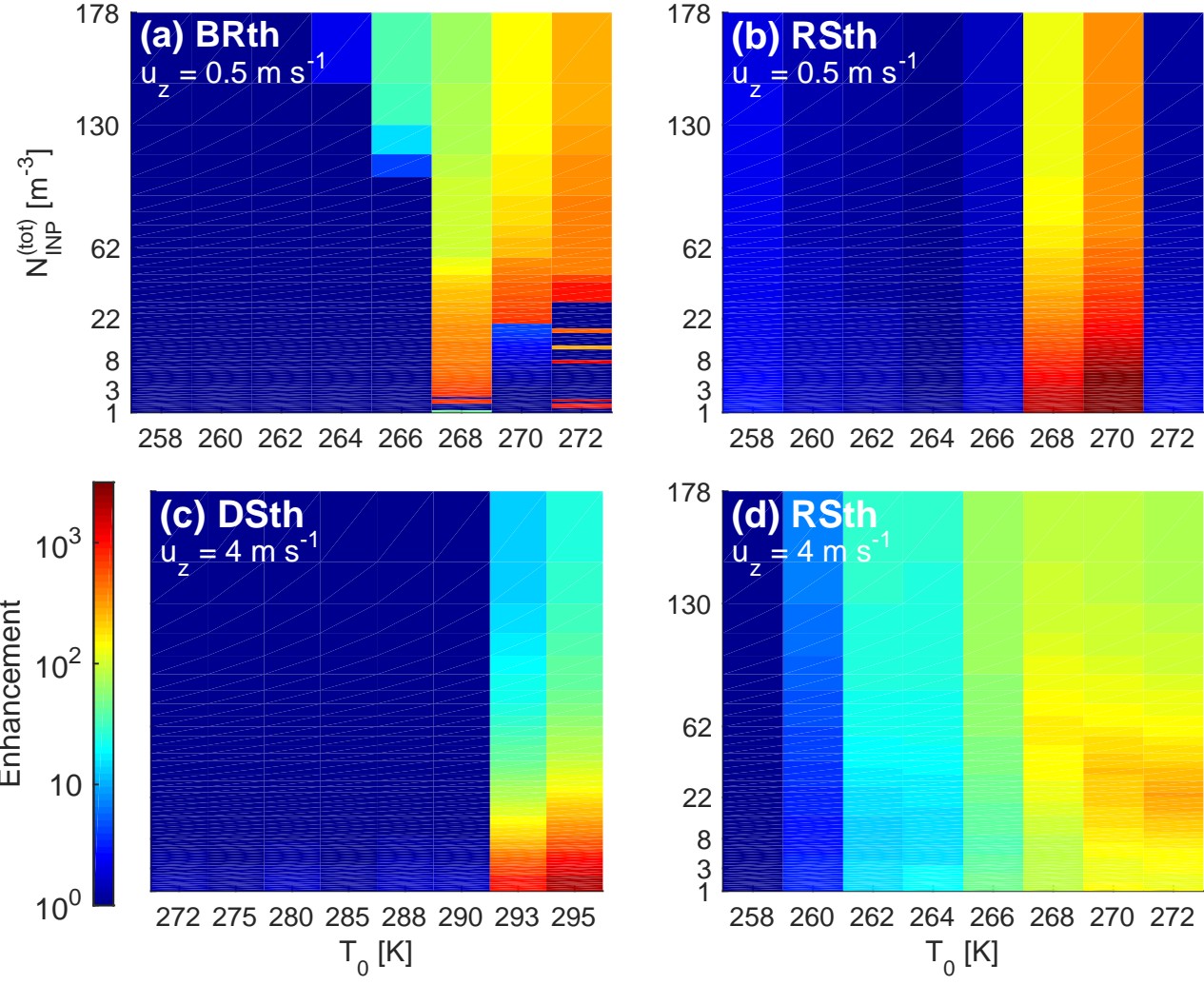

**Figure 3.** ICNC enhancement, i.e., $N_{\text{ice}}(t_{end})/N_{\text{INP}}(t_{end})$, for the thermodynamics simulations with fixed updraft $u_z$ at various values of the total INP number in the parcel $N_{\text{INP}}^{(tot)}$ and initial temperature $T_0$. Red indicates a larger enhancement per INP. Panels **(a)** and **(b)** show the enhancement for ice-ice collisional breakup and rime splintering at a low, stratiform-like updraft of 0.5 m s$^{-1}$. The lowest updraft of 0.1 m s$^{-1}$ is not shown because only very small enhancements occur. Panels **(c)** and **(d)** show the enhancement for droplet shattering and rime splintering at a higher, convective-like updraft of 4 m s$^{-1}$. No meaningful enhancements are generated by ice-ice collisional breakup at the larger updraft or by droplet shattering at the lower one. Note the different temperature scale for the DSth simulation.

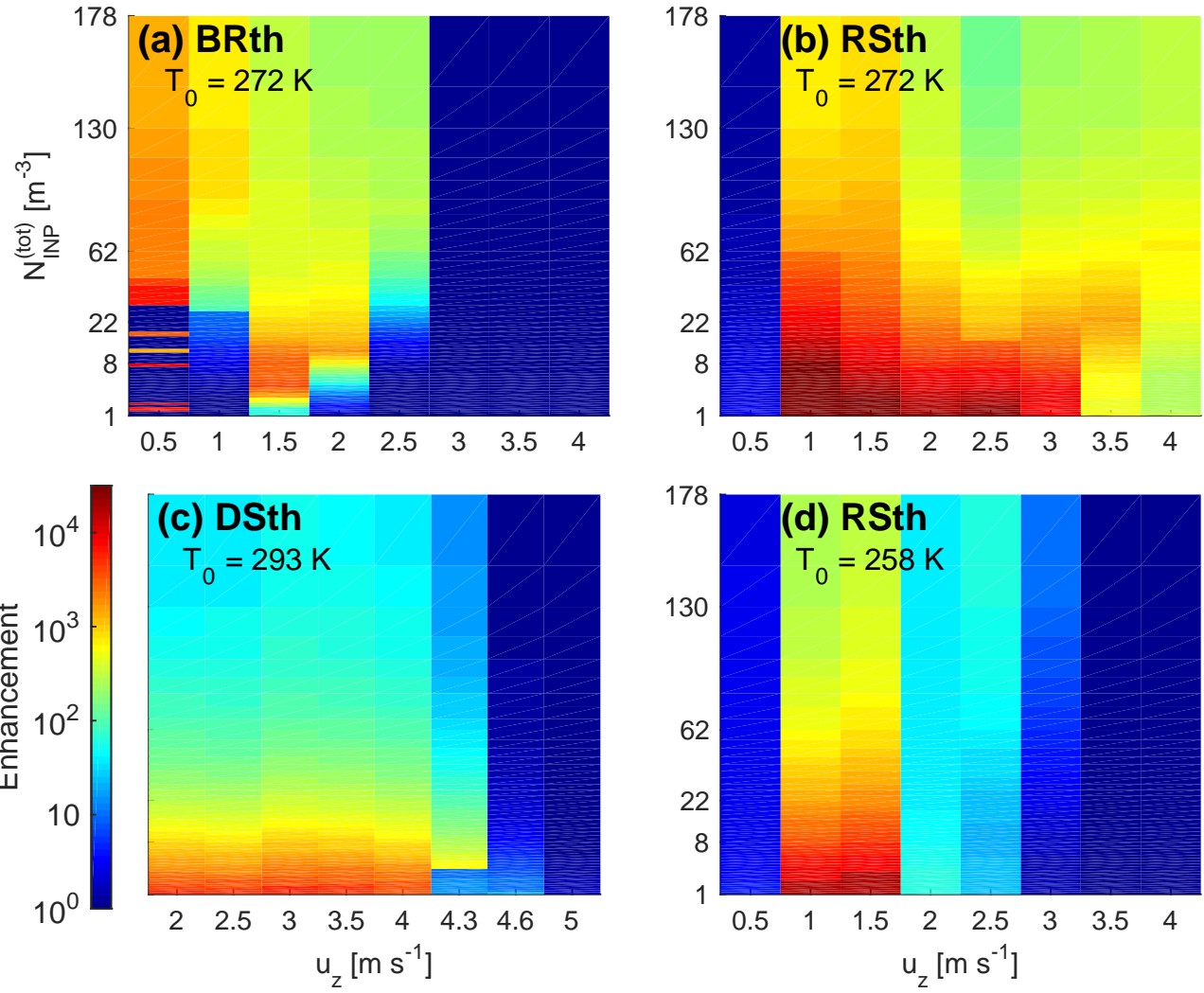

**Figure 4.** ICNC enhancement, i.e., $N_{ice}(t_{end})/N_{INP}(t_{end})$, for the thermodynamics simulations with fixed initial temperature $T_0$ at various values of the total INP number in the parcel $N_{INP}^{(tot)}$ and updraft velocity $u_z$. Red indicates a larger enhancement per INP. Panels **(a)** and **(b)** show the enhancement for ice-ice collisional breakup and rime splintering at a cloud base temperature of 272 K. Panels **(c)** and **(d)** show the enhancement for droplet shattering and rime splintering at colder and warmer cloud base temperatures of 293 and 258 K, respectively. No meaningful enhancement is generated by ice-ice collisional breakup at the warmer $T_0$ or by droplet shattering at the colder ones.

parcel warmer with latent heating. Fragment numbers, $\aleph_{DS}$ and $\aleph_{RS}$, also depend on the large droplet radius or rimed mass, which are reduced at lower supersaturation. Previous work corroborates this understanding: Connolly et al. (2006) found that increasing primary nucleation led to a decrease in the freezing of rain in cloud resolving simulations. Other studies have also emphasized the importance of liquid hydrometeor formation, rather than primary nucleation, to ice generation from rime

splintering (Mossop, 1978, 1985; Hobbs and Rangno, 1985; Heymsfield and Willis, 2014).

The bottom panels show enhancement from rime splintering at a colder $T_0$ and from droplet shattering at a warmer $T_0$. The idea of a 'sweet spot' in $u_z$ reappears: the updraft must be strong enough that large droplets form by coalescence or condensation but modest enough that these droplets remain in an appropriate temperature range to rime or shatter. These trends are summarized in the first panel of Figure 9 and, for rime splintering, agree generally with Mossop (1985) in which

enhancement was possible down to 0.55 m s$^{-1}$ but highest around 1.8 to 2 m s$^{-1}$. Mossop used a shell-fracture hypothesis to explain this optimum: too high a velocity and the riming drop spreads across the ice surface, rather than forming a fragile protuberance, and too small a velocity and an incomplete ice shell may form around the riming drop. Although not a validation of this hypothesis, the simplified model is, interestingly, able to reproduce this $u_z$ behavior without such detailed rime physics.

### 3.3   Parameter perturbations

Lastly we use the insight about $N_{\text{ice}}$ evolution and approximate enhancement from the above simulations to investigate the impact of adjustable parameters. In particular, we look at the effect of generated fragment numbers and temperature dependencies on $N_{\text{INP}}^{(lim)}$ and enhancement magnitude or timing.

First the effect of nucleation rate is investigated on the $N_{\text{INP}}^{(lim)}$ value for breakup, as illustrated in the top panels of Figure 5. Runs are done with a default nucleation rate and ones reduced by factors of 10 and 100, and the conditions for which no

enhancement occurs are shown in black. The number of these points increases dramatically as the nucleation rate decreases from left to right (8 to 32 to 84%). Then as $T_{min}$ increases, the temperature range over which ice-ice collisional breakup occurs shrinks, and $N_{\text{INP}}^{(lim)}$ increases: more ice crystals are needed initially to reach a 100-fold enhancement ultimately. As $F_{BR}$ increases, more fragments are formed per collision, and $N_{\text{INP}}^{(lim)}$ decreases. This second effect of $F_{BR}$ is the larger of the two. These $N_{\text{INP}}^{(lim)}$ trends for ice-ice collisional breakup occur until a sufficiently low $F_{BR}$ or sufficiently high $T_{min}$, beyond

which enhancement does not occur for any value of $N_{\text{INP}}^{(tot)}$ (up to 300 L$^{-1}$).

The bottom panels show $N_{\text{ice}}$ evolution for various values of $F_{BR}$ and $T_{min}$ and for $N_{\text{INP}}^{(tot)}$ of 0.0129 L$^{-1}$ (in yellow) and 0.167 L$^{-1}$ (in green). The effect of both parameters is much larger when $N_{\text{INP}}^{(tot)}$ is small. Increasing $F_{BR}$ from 40 to 280 increases $N_{\text{ice}}$ by a factor of 200 when $N_{\text{INP}}^{(tot)}$ is 0.0129 L$^{-1}$ and by only a factor of 3 when $N_{\text{INP}}^{(tot)}$ is 0.167 L$^{-1}$. Similarly, decreasing $T_{min}$ from 258 to 246 K increases $N_{\text{ice}}$ by a factor of 230 when $N_{\text{INP}}^{(tot)}$ is 0.0129 L$^{-1}$ and by only a factor of 1.5

when $N_{\text{INP}}^{(tot)}$ is 0.167 L$^{-1}$. The parameters also mostly affect the enhancement magnitude not its timing.

Then the effect of shattering probability and generated fragment number are investigated for droplet shattering. We triple the leading coefficient $F_{DS}$ and alter the diameter dependence from quartic to cubic within the Lawson et al. (2015) formulation. We also use two sigmoids shown in Figure S3, which generate higher $\aleph_{DS}$ at small $D_R$ and lower $\aleph_{DS}$ at large $D_R$ relative to

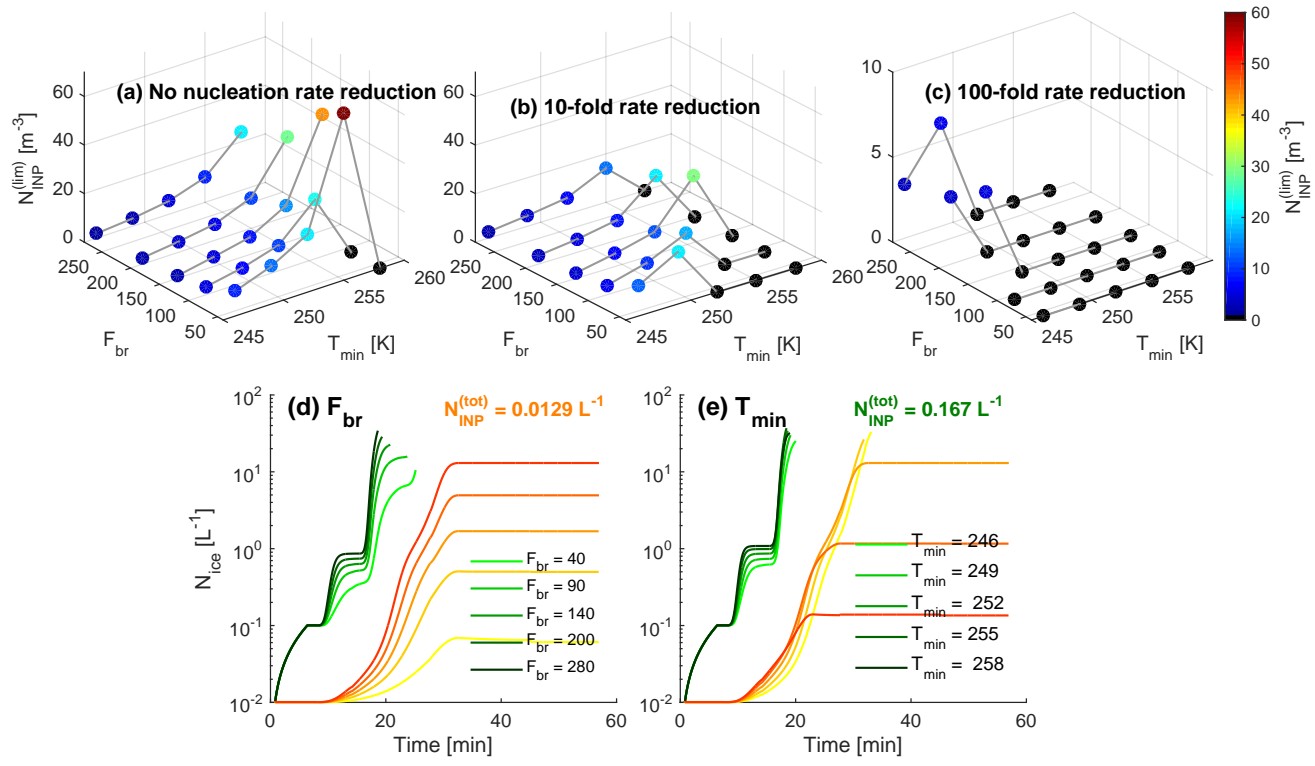

**Figure 5.** Results from the parameter perturbation simulations with ice-ice collisional breakup. The top panels show $N_{\text{INP}}^{(lim)}$ to obtain a 100-fold ICNC enhancement for various values of $F_{BR}$ and $T_{min}$ within the ice-ice collisional breakup parameterization. Dots are also colored by $N_{\text{INP}}^{(lim)}$, where black indicates no 100-fold enhancement ever occurring. From panel **(a)** to **(b)** to **(c)**, the nucleation rate decreases by two orders of magnitude; note that the y-axis in panel **(c)** has a smaller range than the others. The bottom panels show the temporal evolution of $N_{\text{ice}}$ for the various values of $F_{BR}$ and $T_{min}$ with $N_{\text{INP}}^{(tot)}$ of 0.167 L$^{-1}$ (green traces) and 0.012 L$^{-1}$ (yellow traces). The light-to-dark gradient in green and yellow corresponds to the same parameter values. These parameter perturbations are run for $u_z$ of 2 m s$^{-1}$ and $T_0$ of 272 K.

Lawson et al. (2015), based upon the results of droplet levitation experiments. As above, there is no meaningful $N_{\mathrm{INP}}^{(lim)}$ here, so we focus on the maximum enhancement from these various cases, shown in Figure 6.

In panel **(a)**, by far the smallest enhancement occurs for a $D_R^3$ dependence in $\aleph_{DS}$. Independent of $p_{sh}^{(max)}$ these simulations never produce an ICNC enhancement greater than about 50. The largest enhancement comes from sigmoidal dependence on $D_R$ as this yields higher fragment numbers than the polynomial dependence for droplets of less than about 350 $\mu$m diameter, which dominate in our simulation. The situation would be reversed for very large droplets with millmeter diameters: the polynomial function of $D_R$ would predict higher fragment numbers than the sigmoidal one. In all cases, increasing $p_{sh,}^{(max)}$ increases enhancement with a similar, linear effect throughout its range as expected: a 2-fold increase of $p_{sh}^{(max)}$ from 5 to 10% has the same quantitative impact as a 2-fold increase from 10 to 20%.

Panels **(b)** and **(c)** show $N_{\mathrm{ice}}$ evolution for various values of $p_{sh}^{(max)}$ and the sigmoidal and default $D_R^4$ and $\aleph_{DS}$ forms respectively. The yellow traces show this evolution for $N_{\mathrm{INP}}^{(tot)}$ of 0.0129 L$^{-1}$ and the green for 0.167 L$^{-1}$, but these INP concentrations do not make a significant difference. This evolution confirms that the sigmoidal $\aleph_{DS}$ calculates more fragments than the polynomial one: $N_{\mathrm{ice}}^{(max)}$ is 143.2 L$^{-1}$ in **(b)** and 42.1 L$^{-1}$ in **(c)**. And increasing $p_{sh}^{(max)}$ by a factor of 10 from 1 to 10% translates linearly to a factor 10 increase in $N_{\mathrm{ice}}^{(max)}$.

Finally, we investigate the impact of the fragment number from rime splintering, $F_{RS}$. Here we consider enhancement timing because the thermodynamic simulations show that there is no meaningful $N_{\mathrm{INP}}^{(lim)}$ and the default ones show that the enhancement magnitude stays more or less constant. Panel **(a)** shows how the enhancement timing varies with the nucleation rate and fragment number $F_{RS}$. Slower nucleation rates are quantified by a reduction factor $f_{red}$ on the y-axis. Along with lower $F_{RS}$, slower nucleation yields longer enhancement times, by about 8 minutes relative to the highest nucleation rate and $F_{RS}$. $F_{RS}$ is the more influential factor in timing. Its impact on $N_{\mathrm{ice}}$ evolution is shown in panel **(b)**, where a given enhancement is obtained over a shorter period for a higher $F_{RS}$. As for ice-ice collisional breakup, the effect of the parameter is much larger when $N_{\mathrm{INP}}^{(tot)}$ is smaller in the yellow traces.

## 4   Observational comparison and discussion

In the first set of simulations, we investigated $N_{\mathrm{ice}}$ temporal evolution. For breakup and rime splintering at the higher values of $N_{\mathrm{INP}}^{(lim)}$, 100-fold enhancement is formed within 15 to 25 minutes. These values are in agreement with the time scales measured during the COnvective Precipitation Experiment: no single process was definitely active, but drizzle drops formed in 20 minutes and glaciation occurred in 12 to 15 minutes after first ice nucleation (Taylor et al., 2016). In observations of maritime cumuliform clouds, Hobbs and Rangno (1990) measured 100-fold increases in $N_{\mathrm{ice}}$ over 9 minutes. These clouds had tops no colder than -8°C, so rime splintering was the active process, and a similar ice formation rate can be seen in Figure 1c between about 20 and 30 minutes.

These rates are somewhat faster than those measured during the Ice in Clouds Experiment-Tropical campaign in which ice crystal concentrations of about 10 L$^{-1}$ were formed over half an hour Heymsfield and Willis (2014). The model also predicts somewhat faster rates than those in some laboratory studies. For example, Vardiman (1978) measured a 10-fold increase in ice

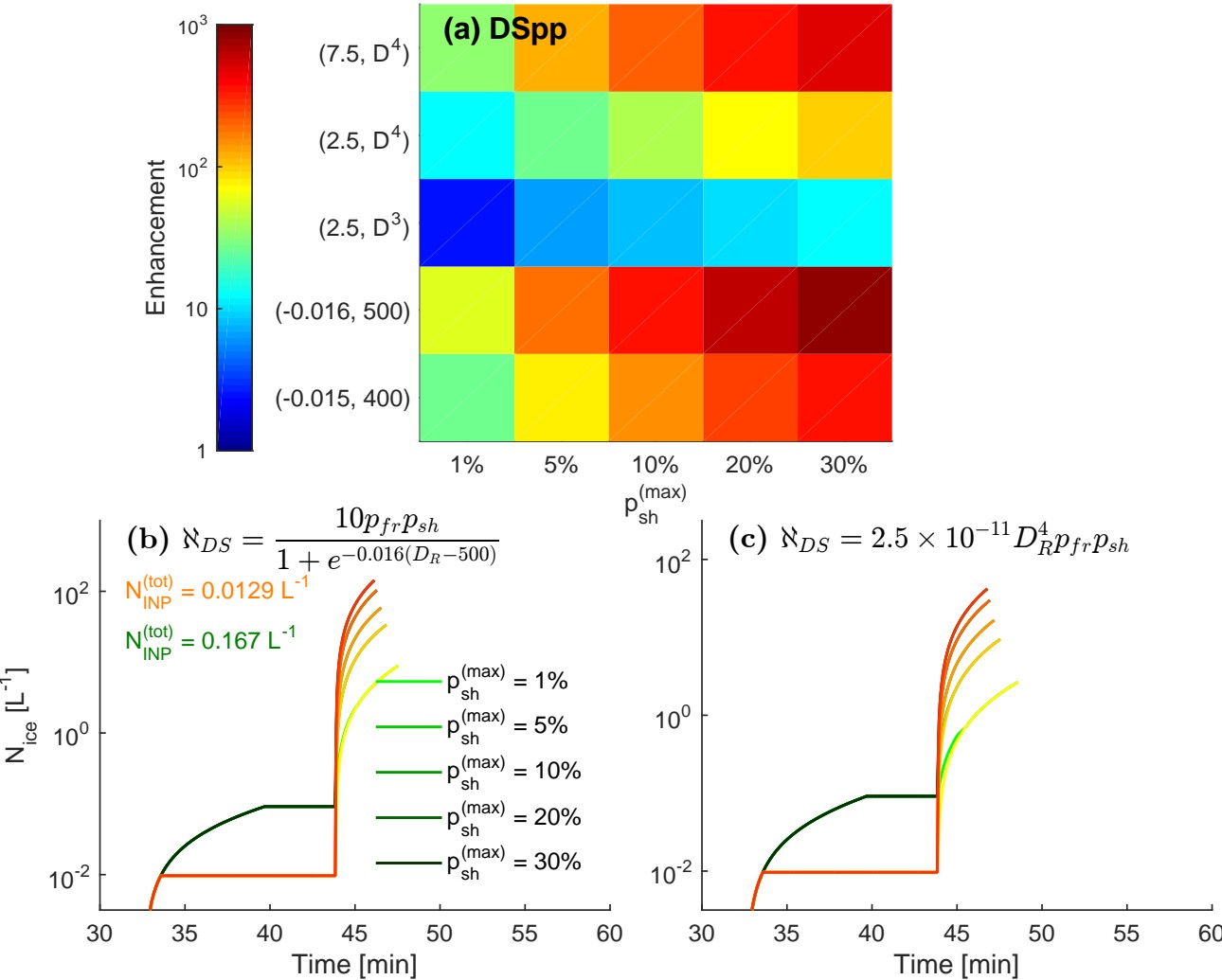

**Figure 6.** Results from the parameter perturbation simulations with droplet shattering. Panel **(a)** shows how the enhancement magnitude shifts with $p_{sh}^{(max)}$ and the various fragmentation functions, both polynomial and sigmoidal. Panels **(b)** and **(c)** show the temporal evolution of $N_{ice}$ for the various values of $p_{sh}^{(max)}$ with $N_{INP}^{(tot)}$ of 0.167 L$^{-1}$ (green traces) and 0.012 L$^{-1}$ (yellow traces) and for a sigmoidal and polynomial fragmentation function respectively. These parameter perturbations are run for $u_z$ of 2 m s$^{-1}$ and $T_0$ of 293 K.

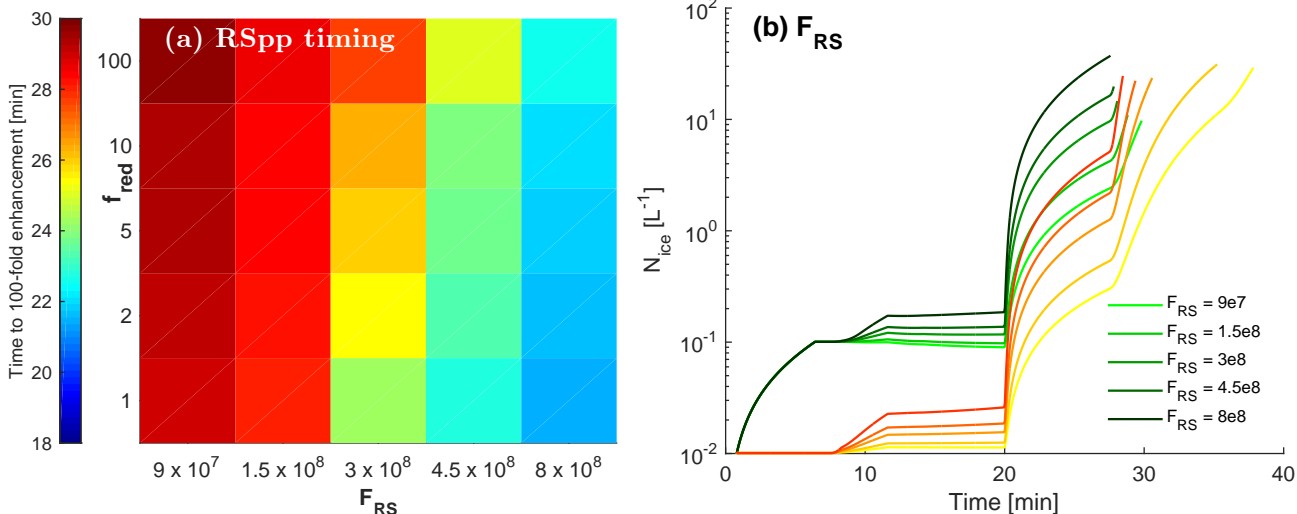

**Figure 7.** Results from the parameter perturbation simulations with rime splintering. Panel **(a)** shows how time of a 100-fold enhancement shifts with the fragment number per kilogram rime $F_{RS}$ and the nucleation reduction rate $f_{red}$. Panel **(b)** shows the temporal evolution of $N_{ice}$ for various values of $F_{RS}$ with $N_{INP}^{(tot)}$ of 0.167 L$^{-1}$ (green traces) and 0.012 L$^{-1}$ (yellow traces). The light-to-dark gradient in green and yellow corresponds to the same parameter values. These parameter perturbations are run for $u_z$ of 2 m s$^{-1}$ and $T_0$ of 272 K.

crystal number over 20 minutes with $N_{INP}^{(lim)}$ of 3 L$^{-1}$. The breakup simulation with $N_{INP}^{(lim)}$ of 2.15 L$^{-1}$ (Fig. 1a) shows must faster evolution. Whereas Hallett and Mossop (1974) measured a linear increase between 10 and 20 minutes, Figure 1c shows exponential increase. These larger rates are due, in part, to the Lagrangian nature of the simulations. By following a single parcel of air, we track colder and colder temperatures and ice accumulation, which both lead to more efficient ice generation.

We have also considered the impact of varying thermodynamics, particularly cloud base temperature and updraft, and INP concentrations. In-situ measurements of breakup are hard to definitively obtain, but our simulations confirm the strong modulation of ultimate $N_{ice}$ by the initial crystal concentration, reported in the laboratory experiments of Vardiman (1978) (his Figure 1). Rangno and Hobbs (2001) also saw about 47% fragmented ice in their measurements of supercooled Arctic clouds, whose cloud base temperatures were around 1 °C and top temperatures around -15 to -18°C. These conditions were appropriate for

breakup and associated with a pristine columnar ice concentration of 0.1 to 3 L$^{-1}$. These values fall in the range of $N_{INP}^{(tot)}$ for which we predict active breakup in Figure 1b.

      Then for processes involving the liquid phase, observations indicate low values of $N_{INP}^{(lim)}$. For example, Crawford et al. (2012) note an $N_{INP}^{(lim)}$ of 0.01 L$^{-1}$ for rime splintering, while Lawson et al. (2015) report INP concentrations of 10$^{-4}$ to 0.01 L$^{-1}$ prior to secondary enhancement. Beard (1992) notes an $N_{INP}^{(lim)}$ of 1 m$^{-3}$ in his measurements of warm-base convective

clouds. The essentially negligible values of $N_{INP}^{(lim)}$ from our simulations reflect these measurements qualitatively. Quantitatively, the simulated estimates may be lower because the model does not represent continuous sedimentation or advection of existing ice outside the parcel.

For rime splintering, both our simulations and observations show the favorability of modest $u_z$ for rime splintering Heymsfield and Willis (2014). Moderate updrafts hold the hydrometeors in the appropriate temperature zone for a longer period of time. On the other hand for droplet shattering, increasing $u_z$ accelerates droplet growth by condensation or coalescence and enhancement, at least up to a certain point, in our simulations. Lawson et al. (2015) also observed the highest $N_{ice}$ enhancement in high-updraft convective cores during ICE-T. In updating model formulations, we have also found that a representation of droplet size distribution broadening is a crucial factor to reproduce behavior from recent observational studies (e.g. Taylor et al., 2016; Lawson et al., 2017). Without a basic implementation of the warm rain process, our model does not give realistic dependence of $N_{ice}$ on $T_0$. Unrealistic coalescence rates in a monodisperse-class hydrometeor scheme should lead to underestimations of secondary ice production, since the largest-diameter droplets are omitted. This reflects an important limitation of the model in its assumption of monodispersity. Large droplets shatter more effectively, and large graupel will have a larger sweep-out kernel for ice-ice collisional breakup. As more observational constraints become available, secondary production processes should be implemented into more complete microphysics schemes to quanitfy the importance of hydrometeor size distribution tails to their tendencies.

Other more advanced features of a real-world parcel, like ventilation effects, spatial phase separation, and continuous sedimentation, could also alter the simulation results (Sullivan et al., 2017). For example, droplet or ice hydrometeor growth will be enhanced by the stronger vapor density gradient generated by their in-cloud motion. Omitting additional hydrometeor growth should again underestimate secondary production. If 'pockets' of ice phase exist within mixed-phase cloud, then the values of $N_{INP}^{(lim)}$ will be more influential as the ice-ice collisional breakup contribution will increase relative to rime splintering or droplet shattering. If a continuous formulation of sedimentation were substituted for the threshold one used here, the largest enhancement in Figures 3 and 4 should shift to higher updrafts. Large hydrometeors would be held aloft by these higher updrafts and feed into the secondary production tendencies.

## 5  Summary and Outlook

We have performed three sets of simulations with a six hydrometeor class parcel model, considering the effect of thermodynamics and parameter perturbations on $N_{INP}^{(lim)}$, as well as ICNC enhancement and timing. Our findings can be summarized in three points:

1. *The evolution of $N_{ice}$ from secondary production is determined by collision-based non-linearity and single versus two-phasedness.*

   $N_{ice}$ increases gradually for the collision-based processes of breakup and rime splintering, whereas for non-collisional droplet shattering, $N_{ice}$ increases abruptly, when $p_{fr}$ and $p_{sh}$ become large enough at cold enough temperatures. $N_{INP}^{(tot)}$ affects both the enhancement magnitude and timing for ice-ice collisional breakup. For rime splintering, $N_{INP}^{(tot)}$ affects timing to obtain a given $N_{ice}(t_{end})$, while for droplet shattering, it has almost no impact on either magnitude or timing. These trends are summarized qualitatively in Figure 8a.

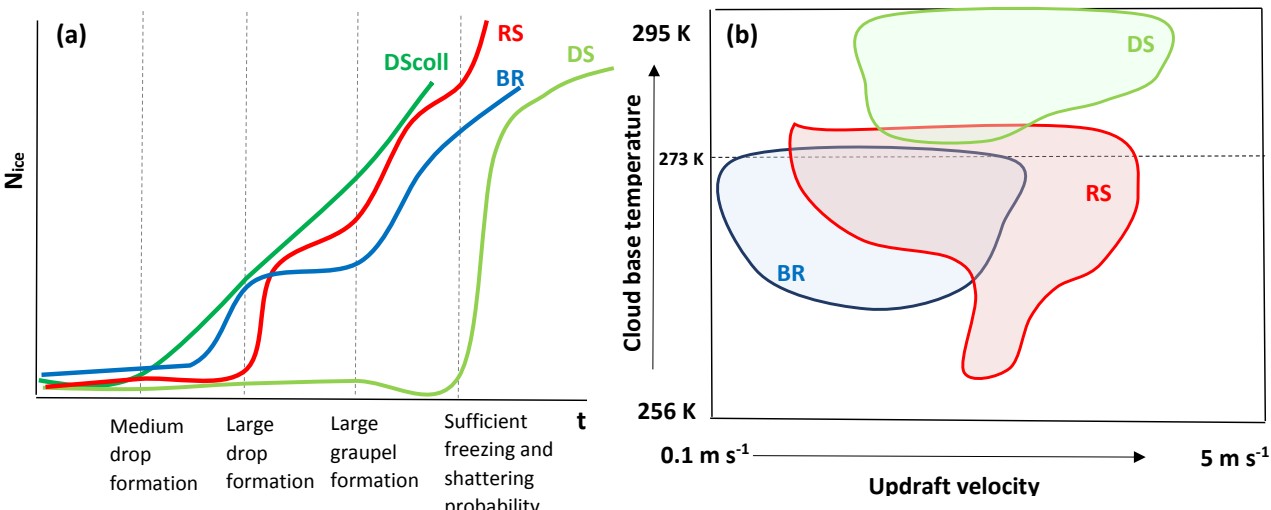

**Figure 8.** Qualitative summary of the findings from the default and varying thermodynamic simualtions in Section 3.1 and 3.2. Panel **(a)** summarizes $N_{\text{ice}}$ evolution for the different processes, particularly the instances of influential hydrometeor formation and whether the process exhibits gradual or threshold increases. Panel **(b)** shows which processes are possible for conditions in the $T_0$-$u_z$ space.

2. $N_{INP}^{(lim)}$ *can be as large as 0.15 L$^{-1}$ for ice-ice collisional breakup. Rime splintering or droplet shattering enhancement is determined by a thermodynamic 'sweet spot' rather than by $N_{INP}^{(lim)}$.*

   $N_{\text{INP}}^{(lim)}$ increases for ice-ice collisional breakup as the fragment number decreases or the temperature range shrinks, particularly for $N_{\text{INP}}^{(tot)}$ of 0.01 L$^{-1}$ or less. At faster nucleation rates, the fragment number and temperature range are also more influential: enhancement occurs for 90% of the parameter space at a default nucleation rate, and just 10% of the space at a rate 100 times slower. These trends are visualized in the 'primary ice' panel of the summary schematic (Fig. 9).

   For rime splintering or droplet shattering, ICNC enhancement of $10^2$ or $10^3$ is possible even for slow nucleation rates and $N_{\text{INP}}^{(tot)}$ as low as 1 m$^{-3}$. For these processes involving the liquid phase, an intermediate updraft for which hydrometeors grow fast enough but also spend long enough in the appropriate temperature zone is more important. For droplet shattering, a representation of the warm rain process and a warm enough initial temperature are also crucial to reproduce observations. These trends are summarized visually in Figure 8b.

3. *No single secondary ice production process dominates ICNC enhancement.*

   At higher nucleation rates, low $u_z$, and warm $T_0$, the contribution from ice-ice collisional breakup is large. If INP are limited, $u_z$ is somewhat higher and $T_0$ is above the freezing level, droplet shattering is most important. And if temperature

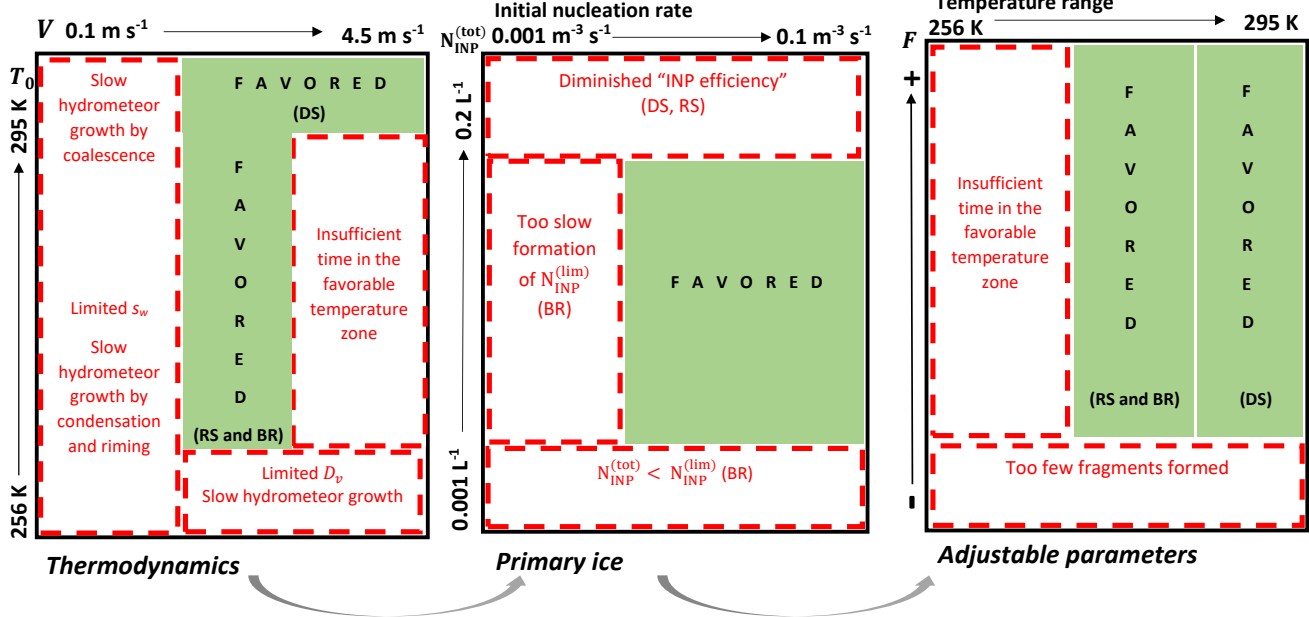

**Figure 9.** Summary of thermodynamic, primary ice, and adjustable parameter trends affecting ICNC enhancement from secondary production. $F$ denotes the leading coefficient of a fragment number function for process $X$, $\aleph_X$. Regions in red indicate that secondary production may be limited, and those in green indicate that conditions are favorable. If the limitation is applicable only to one process, this is indicated in parentheses. The INP efficiency mentioned in the primary ice panel refers to the idea that lower secondary enhancement per INP is produced as the INP concentration increases.

falls around the optimal zone of 268 to 270 K with an intermediate $u_z$, the rime splintering contribution will be large. These "thermodynamic spaces" where one process dominates are visualized in Figure 8b.

More generally, the role of ice-nucleating particles in secondary production reflects how changing aerosol emissions will affect cloud phase partitioning. The low or non-existent values of $N_{INP}^{(lim)}$ calculated in this study indicate that perturbations
5   in CCN concentrations are more influential on mixed-phase partitioning than those in INP concentrations, with the caveat that thermodynamic conditions are appropriate for secondary production. If the mixed-phase cloud is polluted by more CCN, the higher droplet number will mean that fewer droplets reach a sufficient size to shatter or rime efficiently (This last factor has been called the riming indirect effect (Borys et al., 2003; Lance et al., 2011; Lohmann, 2017)). And in these cases, the supercooled liquid fraction remains higher, and the cloud reflects more shortwave radiation. More pollution by CCN could also
10   yield a thermodynamic indirect effect in which latent heat is released at high altitudes and strengthens the upward movement of the cloud; Koren et al. (2005) have called this cloud invigoration. Our simulations have shown that beyond a certain updraft, secondary production is no longer favored. In this way, the liquid portion of a mixed-phase cloud could also remain higher.

The impact of INP concentrations could be larger for deep convective clouds in which anvil spreading is caused by generation of many small crystals at cloud top (Fan et al., 2013). If the cloud is polluted by more INP, more vigorous secondary production by ice-ice collisional breakup may occur under conditions of fast enough nucleation rate but modest enough updraft and warmer subzero cloud base temperatures. These conditions can be found in deep convective clouds, for example at the edges of rising

turrets or tops of eroding ones (Beard, 1992). In contrast to the riming or thermodynamic indirect effects mentioned above, an ICNC increase at the deep convective cloud top, a kind of 'anvil enhancement effect', would radiatively warm the surface.

A systematic quantification of $N_{\mathrm{INP}}^{(lim)}$ is also relevant for the growing field of bioaerosol. Primary biological aerosol particles (PBAP) exist in the atmosphere at much lower number concentrations than dust or black carbon. But they also nucleate at warmer subzero temperatures (Hoose and Möhler, 2012; Fröhlich-Nowoisky et al., 2016), and small biological residues can

intermix with dust particles to boost ice nucleation activity (Conen et al., 2011; O'Sullivan et al., 2015; Steinke et al., 2016). Even when their contribution to primarily nucleated ICNC is small, they may remain influential via initiation of secondary ice production. For example, the ice active fraction of $10^{-4}$ for *Pseudomonas syringae* measured by Möhler et al. (2008) around -8°C could provide the $0.01\,\mathrm{L}^{-1}$ seed concentration from Crawford et al. (2012) for concentrations of $10^5\,\mathrm{m}^{-3}$, although this is an upper bound for bioaerosol number. From our calculations, it could also provide the $N_{\mathrm{INP}}^{(lim)}$ necessary for ice-ice collisional

breakup to occur. Bioaerosol could also be sufficient to initiate rime splintering, given that this process occurs even for $N_{\mathrm{INP}}$ below $1\,\mathrm{m}^{-3}$ in our simulations. A climatically important linkage has also been hypothesized between PBAP, in-cloud ICNC, and cold phase-initiated rain and is often termed the 'bioprecipitation feedback' (Huffman et al., 2013; Morris et al., 2014). The possibility of secondary production with a low $N_{\mathrm{INP}}^{(lim)}$ means that even a few bioaerosol could trigger generation of many small ice hydrometeors from larger droplets or graupel and suppress precipitation.

As a summary of our findings, we present an organizational framework for future studies of secondary production in Figure 9. Favorable conditions for large ICNC enhancement are shown in green, e.g., intermediate updraft in the thermodynamic panel or higher nucleation rate for ice-ice collisional breakup in the primary ice panel. This classification, along with the $T_0 - u_z$ space in Figure 8b, can be used to determine where signatures of secondary production are likely to be found in in-situ or remote sensing data. And as more experimental studies to quantify the fragment number and temperature dependencies of

these processes are done, more quantitative bounds can be established in the final adjustable parameter panel.

*Code availability.*   No data was used in producing this manuscript. Various model version codes are available upon request.

**Appendix:  Notation**

$a_X$   Spheroidal axis of hydrometeor of type $X$

$\beta$   Adjustable parameter in the sigmoidal function for the fragment number generated from shattering

$c_0$   Primary ice nucleation rate based upon DeMott et al. (2010)

$\mathcal{D}_v$ Diffusion coefficient of water vapor

$F_{BR}$ Leading coefficient of the fragment number generated per collision based upon data from Takahashi et al. (1995)

$F_{DS}$ Leading coefficient of the fragment number generated per shattering droplet as in Lawson et al. (2015)

$f_{imm}$ Fraction of INP that immediately immersion nucleate droplets, rather than later after coalescence

$f_{red}$ Factor for nucleation rate reduction

$F_{RS}$ Leading coefficient of the fragment number per kilogram of rime as in Hallett and Mossop (1974)

$\gamma$ Adjustable parameter in the sigmoidal function for the fragment number generated from shattering

$\Gamma$ Atmospheric lapse rate

**ICNC** In-cloud ice crystal number concentration

**INP** Ice-nucleating particle

$K_X$ Gravitational collection kernel for process $X$

$\aleph_{BR}$ Fragment number from ice-ice collisional breakup per large and small graupel number

$N_d$ Small droplet number concentration in the parcel

$\aleph_{DS}$ Fragment number from droplet shattering per large droplet number

$\aleph_{DS}^{(coll)}$ Fragment number from collisional droplet shattering per large droplet and small ice crystal number

$N_i$ Ice crystal number concentration in the parcel

$N_{\text{ice}}$ Total ice hydrometeor number within the parcel, i.e., the summation of ice crystal, small and large graupel numbers

$N_{\text{ice}}^{(max)}$ Maximum $N_{\text{ice}}$ formed within the parcel during a given simulation

$N_{\text{INP}}^{(lim)}$ Limiting ice nucleating particle number concentration to initiate secondary production

$N_{\text{INP}}^{(tot)}$ Total number of ice nucleating particles within the parcel available for primary nucleation. This value is fixed by the user beforehand.

$N_g$ Small graupel number concentration in the parcel

$N_G$ Large graupel number concentration in the parcel

$N_r$ Medium droplet number concentration in the parcel

$N_R$  Large droplet number concentration in the parcel

$\aleph_{RS}$  Fragment number from rime splintering per large droplet and large or small graupel number

$\rho_w$  Density of liquid water

$p_{fr}(t,T,r)$  Temperature- and INP-dependent probability that a large droplet freezes based upon Paukert et al. (2017)

$p_{sh}(T)$  Temperature-dependent probability that a frozen large droplet shatters with $p_{sh}^{(max)}$ being the maximum of this distribution

$r_X$  Radius of hydrometeor of type $X$

$s_w$  Supersaturation with respect to liquid water in the parcel

$\tau_X$  Time delay for a hydrometeor in class $X$ to grow by deposition, riming, or condensation to the next class

$T_0$  Cloud base temperature or the initial temperature of the parcel

$t_{end}$  Time when the simulation is terminated, either because the parcel has become water subsaturated or the temperature has reached 237 K where homogeneous nucleation can occur

$T_{min}$  Minimum temperature for ice-ice collisional breakup to occur

$u_z$  Updraft velocity of the parcel

*Competing interests.*  The authors declare no competing interests.

*Acknowledgements.*  S.C.S. and A.N. acknowledge funding from a NASA Earth and Space Science Fellowship (NNX13AN74H), a NASA MAP grant (NNX13AP63G), and a DOE EaSM grant (SC0007145). A.N. acknowledges funding by the European Research Council Consolidator Grant 726165 (PyroTRACH). C.H. acknowledges funding by the Helmholtz Association through the President's Initiative and Networking Fund (VH-NG-620) and by the Deutsche Forschungsgemeinschaft (DFG) through projects HO4612/1-1 and HO4612/1-2. No

data was used in producing this manuscript. Various model version codes are available upon request.

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
