# Peer review of "Initiation of secondary ice production in clouds"

_Atmospheric Chemistry and Physics, 2017_

## Referee Comment (RC1) · Anonymous Referee #1 · 9 Jun 2017

MAJOR COMMENTS

I will start this review by confessing that I am an observationalist, not a modeler. I bring an obvious bias into this review, which is that I couch my evaluation of this work in terms of data collected in clouds, not numerical simulations of clouds. My main concern with this manuscript is that I cannot determine how this parcel model relates to an updraft in a cloud. Presumably, a parcel model is intended to represent the evolution of an undiluted parcel of cloud as it rises in the atmosphere. However, it is not clear what the prognostic microphysical variables are in the model. Presumably the model predicts $N_{ice}$ for each of the categories because this is shown in the "ice generation function" equation, but what about mass? It's not even clearly stated if it's a bulk or bin microphysics scheme. It's also not clear what the "ice generation function" itself is, and what the units of $G_{ice}$ are. I'm assuming this is $dN_{ice}/dt$ from all microphysical processes, but it's not clear. The Sullivan (2017 – JGR) reference is in review and of no help. Even if the Sullivan JGR paper becomes available, at a minimum the manuscript should state what the model predicted variables are, and how they are being solved in the model numerically (e.g., what kind of time stepping method, the time step, etc.). It would also help if the manuscript gave the evolution equations for the model predicted variables.

The manuscript shows no drop or ice particle size distributions and no liquid water or ice water contents as a function of temperature. Also, the observations that I am most familiar with suggest that clouds with cloud-base temperatures colder or equal to 0 C, which are all of the cases examined here, do not produce cloud drops large enough to support drop shattering, and generally not even large enough to support rime splintering. Large drops (drizzle and rain drops) are what the literature (e.g., Koenig 1963, 1965; Hobbs and Rangno 1990, Rangno 2008, Lawson et al. 2015) associates with drop shattering and rapid glaciation. The data suggest that the formation of millimeter-diameter supercooled drops requires cloud base temperatures warmer than approximately +18 C (291 K) and broad (> 50 µm diameter) cloud base drop distributions. Albeit, the requisite relationship between CCN and cloud base temperature is yet to be accurately quantified. Also, the coalescence process is key to the formation of supercooled large drops. Nowhere in the manuscript can I find how coalescence is handled in the model (except that $K_x$ is a gravitational collection kernel in Eq. 1). One aspect of the simulations that does appear to be consistent with the observations is that rime-splintering takes place only in clouds with very weak updrafts (e.g., Heymsfield and Willis 2014). However, it is not clear in the manuscript exactly why this takes place in the simulations. Before I can recommend publication, the manuscript needs to provide an explicit description of the model, and the evolution of the parcel in terms of microphysical parameters (liquid and water size distributions, LWC, IWC as a function of temperature). I understand that this may be a bit artificial given the six categories of particles, but an attempt must be made, and the results should be compared with observations.

SOME SPECIFIC COMMENTS

Following are some specific comments. Until the major comments are addressed, I am not willing to go through the manuscript with a fine-toothed comb, as I assume that the paper will be significantly modified. The comments below are intended to give the authors some idea of the type of modifications that are needed.

p. 1, Line 4:  "Break Up" is not a good term for ice-ice collisions, because drops also break up.  I suggest that you find a more descriptive term that applies only to ice. If the term "break up" has to be retained, then it needs to be defined as ice-ice collisions here and everywhere else in the manuscript.

p. 2, Line 7:  Add references; there are several.

p. 2, Lines 16-17:  This is contradictory.  In the previous sentence you reference Field as reporting many uncertainties in the physics of secondary ice production, and then go on to state that small-scale models provide a good tool to estimate variability in secondary-produced ice.  The model is only as good as the physics it contains.  With the acknowledged vast degree of uncertainties, how can one have any confidence in the model results?  If the model results are to be useful, then the physical uncertainties have to be emphasized.  Also, sensitivity tests should be run to show how the physical uncertainties impact the results. At a minimum, a disclaimer of this sort needs to be inserted at this point in the manuscript.

p. 3, Eqn (1) and discussion: Eqn (1) is far too arcane to understand what is going on in the model.  The reference to Sullivan et al. (2017) is of no use since it is under review.  There are several unanswered questions.  What are the units of $G_{ice}$?  What is the role of coalescence and how is it handled?  What is the cloud base drop distribution?  Are CCN included?  If so, how?  Why don't small ice and small drops appear in Eqn (1)?  Also, the number of secondary ice particles produced is only one issue.  The mass of ice is of equal if not more importance.  If large (millimeter-diameter) supercooled drops are rapidly freezing, as seen in the observations, then the conversation of water to ice (and eventually back to water in the form of rain), is more significant than the number of ice particles.  Show the results also in terms of water and ice mass.

p. 3, line 27:  237 K is not the homogeneous freezing temperature of pure water.  The generally accepted value in the literature is 235.15 K. The AMS Glossary of Meteorology states that homogeneous nucleation occurs near 233.15 K.

p. 7, Fig. 2 Caption:  How were the values of 2 and 10 fragments per drop chosen?  How is the dependence on drop size handled?

p. 10, Line 17:  Lawson et al. (2015) explicitly states that rime-splintering is not responsible for the observed secondary ice process.  Delete this reference.

p. 11, Lines 4 – 7:  What are the justifications for these assumptions and modifications?

p. 12, Lines 1 – 5:  The production of ice in this scenario may be of some interest, but of more interest to cloud physicists is how the ice and water mass budgets evolve. Please show these.

p. 14, Line 5:  This is the first mention of CCN.  Were CCN used in model, and if so, how?

p. 16, Line 8:  "warm cloud base".  All cloud bases cited in the paper are < 273K, so there are no warm cloud bases.

---

## Referee Comment (RC2) · Anonymous Referee #2 · 26 Jun 2017

The manuscript investigates the role of three secondary ice production mechanisms (rime splintering, frozen droplet shattering, and breakup), more specifically the evolution of the total ice number concentration depending on secondary ice production, the thermodynamic limitations of the secondary processes and the dependence on the chosen parameterization. The authors found that the evolution of the total ice number concentration is determined by the involvement of two phases and the non-linearity of the collision process. However, in case all processes are active none of them is dominant over the others. They also found that only breakup needs a minimum number of ice nuclei, all other processes are more sensitive to the number of cloud condensation nuclei and thermodynamic conditions. The results are summarized in Fig. 8 where they show in which thermodynamic region and also for which ice nuclei concentration

and rates secondary ice formation if favorable. The manuscript adds some interesting aspects to the question what bridges the gap between ice nuclei and ice crystal measurements.

**1 Major comments:**

The simulations show some interesting aspects of secondary ice production. However, the interpretation aspect could be stronger emphasized. What do the findings have for consequences in terms of modeling of mixed-phase clouds? How are the results connected to field observations? Do the findings agree with observations? Do the findings make sense in the general context/understanding of the microphysics of a mixed-phase cloud? Which further aspects would need to be investigated?

**2 Specific comments:**

- page 2, line 23-26: Could you not calculate the number of INP from the nucleation rates?

- page 3, line 4: How do you derive a nucleation rate from the INP concentration given in DeMott et al. 2010?

- page 3, line 4: Why a heaviside function?

- page 3, line 9: How is the connection between $DS$ and $DScoll$? From the description ($DScoll$=collision between large droplet and ice crystals?) it sounds like two different processes. Add more explanation to this point.

- page 3, line 9: Why is it 1% and not 0% outside of the temperature range?

- page 5, line 3-4: Explain that more explicit, example?

- page 7, Eqn. 2: What is the physical concept or idea behind this formula or approach?

- page 9, line 2: Where do I see that in the figure?

- page 9, line 9: Again: where and how do I see that in the figure?

- page 10, line 14-17: What is the reason for these differences?

- Section 3.3: Make it clear what process which paragraph is referring to, it starts with Break up, page 11, line 4 DS...

- page 13, line 10: Please also describe what can be seen in panel (b).

- page 14, line 1: You did not really explain the single versus two-phasedness before? It is an interesting aspect and maybe you could explain that a bit further (here or in the sections before).

- page 14, summary point 1: It could be interesting to illustrate this point in a table or figure. In figure 8 it is not really depicted for each process separately.

- page 14, line 23: What do you mean by emissions? Aerosol emissions?

- page 14: It could be interesting to plot the dominant regions of each process on a 2D-Plot with the vertical velocity and the temperature on the axis.

- page 15, line 13: You could add more references here, e.g. Conen et al. 2011, Steinke et al. 2016.

- Figure 1: Panel (d) is not mentioned/explained in the text. Either remove it or explain it in the text as well.

- Figure 2: Explain in Caption what n=2, n=10 means (is only indirectly explained in the text).

- Figure 3: In the case of $BRth$ and $RSth$: does the same argument yield as in $DSth$ for choosing a velocity of 0.5 instead of the smallest value of 0.1?

- Figure 3: Why are there no meaningful enhancements by breakup if the updraft is larger? Less collisions?

- Figure 4: Why are there no meaningful enhancements by breakup at colder $T_0$?

- Figure 8: It is a nice summary of the outcome of the paper and can be quite a useful Figure. You could strengthen that a bit more. In the current version of the manuscript is not very prominent.

- Figure 8: What is meant by diminished INP efficiency?

- Figure S1: You could add $BR$, $RS$ and $DS$.

- Figure S1: The process rime-splintering is not clearly depicted (how does the ice multiplication happen).

**3  Small remarks,typos:**

- The INP subscript is not nice to read, reduce the space between the letters or write it non-italic (which is standard for physical subscripts?).

- page 1, line 20: Year missing at citation Ladino et al..

- page 2, line 28: Delete above.
- page 4, line 3: Replace freezing by melting. 272 K is the melting temperature. Freezing normally happens at lower temperatures.

- page 4, line 6: Also $F_{DS}$ and $F_{RS}$?

- page 4, line 7: The parameters for the functional form are $\beta$ and $\gamma$, if yes add in brackets after "...per shattering droplet"

- page 5, line 13: Remove brackets around citation.

- page 5, line 15: Add "explained in" before Section... .

- page 5, line 25: What does "its magnitude" refer to?

- page 5, line 30: Has to be $N_{ice}^{(max)}$ instead of $N_{INP}^{(max)}$?

- page 7, line 16: Add brackets around the citation "(Paukert et al., 2017)" instead of "Paukert...".

- page 14, line 19: Should or is?

- page 14, line 28: The brackets are strange.

- page 15, line 2: The term "supercooled liquid fraction" might need a sentence of explanation.

- page 16, line 22: This is also a leading coefficient?

- page 17, line 17: $D$ (diameter) is missing in the variable list. Either add it or exchange it here with $r$.

- Table 1: In the Caption there is a run mentioned denoted INP below, which does not exist in the Table?

- Table 1: Reformulate in the Caption (since thermodynamic simulations is ambiguous, $BRth$ etc. is also a thermodynamic simulation): "Thermodynamic simulations run with ... are shown solely ...".

- Table 1: For the simulations only shown in the Supplement ($BRDSth$...) no Table with conditions of the simulations exists. However, this could be helpful in comparison to Table 1.

- Table 1: Run $DSpp$: What does the $D$ mean in the range of values for $F_{DS}$?

- Table S1: $F_{RS}$ and $F_{DS}$: What is frag? fragments?

- Table S1: $p_{sh}$: What is $N$?

- Table S1: $p_{rf}$: What is $A$?

- Legend Figure 5 (a), (b), (c), Fig. 6 (a) and Fig. 7 (a) needs to be bold to be consistent (matters most in case of Fig. 5).

- Figure 5 a, b, c: It is a bit unlucky that most points are in the blueish range of the color scale. It is quite difficult to differentiate the different color tones of blue.

- Figure 5 and 6 and 7: Is the color scale in the legend the same for the green traces and the yellow traces (also only color of green traces is shown)? Mention in the Caption or add the colors for the yellow traces also to the legend.

- Figure 7: The coloring is only similar to Fig. 1 c.) for the first color of the green and yellow traces? I found this comment a bit confusing and it did not add necessary information, so maybe delete it.

- Figure 8: Are the arrows from one panel to the other needed? What do they symbolize?

- Figure 8: What is $s_M$? (left panel)

- Figure 8: What is $D_v$? (left panel)

- Figure S2 Caption: line 1: add on $BRpp$ in the end.

- Figure S2 Caption: line 2: (b) shows the effect of the minimum... function on ... .

- Figure S2 Caption: line 3: ...droplet due to $F_{DS}$?

- Figure S2 Caption: (d) What is plotted here? Freezing probabilities? Does not fit to the plot.

- Figure S3: Replace um with $\mu$m.

- Figure S3 Caption: Add: in dependence of $D_R$.

- Figure S3 Caption: How is the second sentence connected to this figure?

- Figure S4 Caption: Shift bracket behind "number".

- Figure S7: Difficult to read legend (a).

---

## Author Comment (AC1) · 19 Aug 2017

**Responses to Reviewer 1 Comments**

I will start this review by confessing that I am an observationalist, not a modeler. I bring an obvious bias into this review, which is that I couch my evaluation of this work in terms of data collected in clouds, not numerical simulations of clouds. My main concern with this manuscript is that I cannot determine how this parcel model relates to an updraft in a cloud. Presumably, a parcel model is intended to represent the evolution of an undiluted parcel of cloud as it rises in the atmosphere. However, it is not clear what the prognostic microphysical variables are in the model. Presumably the model predicts $N_{ice}$ for each of the categories because this is shown in the "ice generation function" equation, but what about mass? It's not even clearly stated if it's a bulk or bin microphysics scheme. It's also not clear what the "ice generation function" itself is, and what the units of $G_{ice}$ are. I'm assuming this is $dN_{ice}/dt$ from all microphysical processes, but it's not clear. The Sullivan (2017 – JGR) reference is in review and of no help. Even if the Sullivan JGR paper becomes available, at a minimum the manuscript should state what the model predicted variables are, and how they are being solved in the model numerically (e.g., what kind of time stepping method, the time step, etc.). It would also help if the manuscript gave the evolution equations for the model predicted variables.

Thank you for your careful reading and feedback. We initially limited discussion of model development, given that another manuscript treats those details thoroughly. We understand that this manuscript was not yet available upon your first reading, but it has now been accepted (doi:10.1002/2017JD026546). We have also expanded Section 2 to clarify the model description. To more generally make our results in each section relatable to observations, we include the table shown at the end of these responses in our summary and outlook section.

First, the prognostic microphysical variables are the hydrometeor number in each of the six classes. We rewrite: "*The model predicts the number in these six classes, denoted $N_i$, $N_d$, $N_g$, $N_G$, $N_r$, and $N_R$ respectively.*" There is not an $N_{ice}$ "for each of the categories", as you state. Rather, $N_{ice}$ is the summation of the number in the three ice hydrometeor classes. This is stated in the Notation Appendix, and we repeat it for clarity early on in Section 2.

Then, there are no size distributions for the hydrometeors in each class. They are assumed monodisperse but their radii or major axes are evolved over time. To Section 2, we add, "*The hydrometeors in each class are assumed monodisperse, but their sizes are tracked over time as a function of temperature and supersaturation.*" To emphasize that this is a bin microphysics scheme, with six hydrometeor classes, we write "*The **bin** microphysics consists of primary nucleation and …*"

In order to clarify the ice generation function $G_{ice}$, we have rewritten Equation 1:

$$G_{ice} = \frac{dN_i}{dt}\bigg|_{NUC} \quad + \frac{dN_i}{dt}\bigg|_{BR} \quad + \frac{dN_i}{dt}\bigg|_{RS} \quad + \frac{dN_i}{dt}\bigg|_{DS} \tag{1}$$

$$= c_0\, H(t) \quad + \eta_{BR} K_{BR} \aleph_{BR} N_g N_G \quad + \eta_{RS} \aleph_{RS}\Big[K_{RS,g} N_g + K_{RS,G} N_G\Big] N_R \quad + \eta_{DS} \aleph_{DS} N_R \tag{2}$$

The ice generation function includes all sources of ice crystals, i.e., production of ice crystals from primary nucleation, collisional breakup, rime splintering, and frozen droplet shattering. We write this before Equations 1 and 2, along with the fact that the ice generation function has "*units of $m^{-3}\, s^{-1}$*", i.e., it is a number concentration of crystals generated per time. Then we have added an equation for the number balance looks in the ice crystal hydrometeor class:

$$\frac{dN_i}{dt} = G_{ice}(t) - G_{ice}(t - \tau_i) - \eta_{agg} K_{agg} N_i N_g$$

We hope that these adjustments clarify the relationship between the tendencies, generation function, and number balances: the source tendencies make up the generation function, and the generation function at different times makes up the number balance.

Finally we add the following description of the numerical solutions used: "*The six hydrometeor number tendencies are solved with an explicit Runge-Kutta (2,3) pair for delay differential equations [Bogacki 1989] and coupled to moist thermodynamic equations for pressure, temperature, supersaturation, mixing ratios, and hydrometeor sizes. This second set of equations is solved with a Rosenbrock formulat of order 2 [Rosenbrock 1963].*" We address the concern about ice mass below.

The manuscript shows no drop or ice particle size distriutions and no liquid water or ice water contents as a function of temperature. Also, the observations that I am most familiar with suggest that clouds with cloud-base temperatures colder or equal to 0 C, which are all of the cases examined here, do not produce cloud drops large enough to support drop shattering, and generally not even large enough to support rime splintering. Large drops (drizzle and rain drops) are what the literature (e.g., Koenig 1963, 1965; Hobbs and Rangno 1990, Rangno 2008, Lawson et al. 2015) associates with drop shattering and rapid glaciation. The data suggest that the formation of millimeter-diameter supercooled drops requires cloud base temperatures warmer than approximately +18 C (291 K) and broad (> 50 um diameter) cloud base drop distributions. Albeit, the requisite relationship between CCN and cloud base temperature is yet to be accurately quantified. Also, the coalescence process is key to the formation of supercooled large drops. Nowhere in the manuscript can I find how coalescence is handled in the model (except that $K_x$ is a gravitational collection kernel in Eq. 1). One aspect of the simulations that does appear to be consistent with the observations is that rime-splintering takes place only in clouds with very weak updrafts (e.g., Heymsfield and Willis 2014). However, it is not clear in the manuscript exactly why this takes place in the simulations. Before I can recommend publication, the manuscript needs to provide an explicit description of the model, and the evolution of the parcel in terms of microphysical parameters (liquid and water size distributions, LWC, IWC as a function of temperature). I understand that this may be a bit artificial given the six categories of particles, but an attempt must be made, and the results should be compared with observations.

There are no drop or ice particle size distributions shown because the hydrometeors are assumed monodisperse in each class, and we have added this explicitly to the manuscript. For the coalescence formulation, we use a gravitational sweep-out kernel, as you note, and assume that the terminal velocity of the (small) collected droplets is negligible relative to that of the (medium) collector droplets. Then the coalescence efficiency between small and medium droplets is assumed to be unity, and that between droplets of the same size is assumed to be negligible. These assumptions are based upon the measurements of Klett and Davis 1973. We include these detail: "*The coalescence efficiency is assumed to be unity between small and medium droplets and negligible between two droplets of the same size.*"

Your point about the cloud base temperatures is a good one. Indeed, more recent in-situ measurements where droplet shattering was thought to occur cite warmer cloud base temperatures. Given our choice of a less steep CCN spectrum (yielding fewer droplets) and a stronger updraft, large droplet formation should still reasonably occur for colder initial temperatures. As noted in Taylor et al. 2016, "the temperature at which large concentrations of drizzle and raindrops form depends [not only] on the cloud base temperature [but also on] cloud-drop number concentration and time-dependent factors such as updraft speed."

There are also measurements that indicate that the droplets need not be on the order of a millimeter to shatter effectively. Laboratory droplet levitation experiments shown in Leisner et al. 2014 indicate that shattering can occur even for droplets of diameter of 80 um. Lawson et al. 2015 note that droplets about 200 m above the cloud base, at temperatures where freezing and shattering could begin, have

a diameter of 90 um. Nevertheless, we run two sensitivity tests initiated from $T_0$ of 294 K and with a more convective updraft of 5 m s$^{-1}$ and include their $N_{ice}$ evolution in the supplementary material:

[Figure]

In Section 3.1, we note this sensitivity test: "*In more recent measurements with evidence of droplet shattering, the cloud base temperature has been warmer, and the updraft stronger, than the default conditions in Table S1, e.g., Lawson et al. 2015, Taylor et al. 2016 (see Table 2). We show the $N_{ice}$ evolution from a `warm-base-convective' sensitivity run in Figure S7. Here the same threshold behavior occurs once the parcel reaches cold enough temperatures for droplet freezing, but there is no $N_{ice}$ decrease beforehand because ice nucleation begins later, and no graupel has begun to fall out.*" And to a paragraph early on in Section 3.2 on the thermodynamic simulations, we add that "*These cloud base temperatures are colder than those associated with most in-situ measurements of frozen droplet shattering [Lawson et al. 2015, Taylor et al. 2016]; however, our simulations still produce droplets of sufficient diameter to shatter, O (100 um), and a 'warm-base-convective sensitivity run is shown in Figure S7.*"

Finally, we have shown ice crystal numbers because "enhancement" from secondary ice production is generally discussed in the literature in terms of the orders-of-magnitude discrepancy between ice crystal and ice-nucleating particle numbers. But the mass is also important to consider, as you suggest. To the supplemental information, we add the two figures below, ice mass mixing ratio evolution for the default simulations and a sample ice crystal radius evolution. *At the top of Section , we state: The ice mass mixing ratio and ice crystal radius evolution are also shown in Figures S4 and S5, but analysis focuses on $N_{ice}$ below.*"

[Figure]

[Figure]

**SOME SPECIFIC COMMENTS**

Following are some specific comments. Until the major comments are addressed, I am not willing to go through the manuscript with a fine-toothed comb, as I assume that the paper will be significantly modified. The comments below are intended to give the authors some idea of the type of modifications that are needed.

p. 1 Line 4: "Break Up" is not a good term for ice-ice collisions, because drops also break up. I suggest that you find a more descriptive term that applies only to ice. If the term "break up" has to be defined as ice-ice collisions here and everywhere else in the manuscript.
"*Breakup*" was used because preexisting work on this process generally employs this term, e.g., Yano and Phillips *JAS* 2001, Phillips et al. *JAS* 2017, Field et al. *Meteor. Mono.* 2017. But we understand that this terminology may cause confusion with droplet breakup. We have gone through and changed all instances of "*breakup*" to "*collisional breakup*".

p. 2, Line 7: Add references; there are several.
We have added Scott and Hobbs 1977, Phillips et al. 2001, and Fridlind et al. 2007 to the citations for frozen droplet shattering.

p. 2, Lines 16-17: This is contradictory. In the previous sentence, you reference Field as reporting many uncertainties in the physics of secondary ice production, and then go on to state that small-scale models provide a good tool to estimate variability in secondary-produced ice. The model is only as good as the physics it contains. With the acknowledged vast degree of uncertainties, how can one have any confidence in the model results? If the model results are to be useful, then the physical uncertainties have to be emphasized. Also, sensitivity tests should be run to show how the physical uncertainties impact the results. At a minimum, a disclaimer of this sort needs to be inserted at this point in the manuscript.
We do not believe that these statements are contradictory. Investigating how a given output varies with uncertain parameters is an important application of models. And particularly for small-scale, more controllable models, output variation with adjustable parameters can be well-understood. This kind of work allows experimentalists to focus on measuring the most influential parameters and provides a test-bed for parameterizations prior to implementation in large-scale models. This utility of small-scale models is summarized in the IPCC Assessment Report 5: "high-resolution models enhance our

understanding of cloud processes [as] an important tool in testing and improving parameterizations of cloud-controlling processes."

As you note, sensitivity tests should be run with the small-scale model to understand the process and parametric uncertainties. Sections 3.1.1 and 3.3 contain these tests. We run simulations for different formulations of the physics of frozen droplet shattering. And then we investigate the sensitivity to adjustable parameters in the fragment generation functions (particularly $F_{BR}$, $T_{min}$, sigmoid versus polynomial forms for droplet shattering, and $p_{sh}^{(max)}$).

We clarify the utility of small-scale models in this paragraph: "*Laboratory and in-situ data of these processes are difficult to obtain, and their fragment generation functions and temperature dependence remain uncertain [Field et al. 2017]. Given these uncertainties, implementation of secondary ice production parameterization in large-scale models would be premature. Instead, small-scale, more controllable models provide a means of estimating variability in output secondarily-produced ICNC with these parameters, as well as the minimum number of INP needed to initiate secondary production.*"

p. 3, Eqn (1) and discussion: Eqn (1) is far too arcane to understand what is going on in the model. The reference to Sullivan et al. (2017) is of no use since it is under review. There are several unanswered questions. What are the units of $G_{ice}$? What is the role of coalescence and how is it handled? What is the cloud base drop distribution? Are CCN included? If so, how? Why don't small ice and small drops appear in Eqn (1)? Also, the number of secondary ice particles produced is only one issue. The mass of ice is of equal if not more importance. If large (millimeter-diameter) supercooled drops are rapidly freezing, as seen in the observations, then the *conversion* of water to ice (and eventually back to water in the form of rain), is more significant than the number of ice particles. Show the results also in terms of water and ice mass.
As described above, we have worked to make the model description more clear without restating what has already been published in the model development manuscript. In particular, we have more clearly stated the purpose and the units of the ice generation function and expanded its mathematical explanation with two additional equations. Small ice and droplets do not appear in Equation 1 because they play no role in any of the processes that are a source of small ice crystals.

Then we have emphasized that there are no size distributions involved; the monodisperse radius or axis of each hydrometeor class is evolved in time. The model contains no explicit aerosol. We add this statement and an in-line equation for primary nucleation before the statement that "*the droplet generation function consists simply of droplet activation, calculated from a Twomey power-law formulation*." So droplet number is calculated from supersaturation rather than a CCN number. Then we have added more detail for the coalescence formulation to Section 2, as discussed in the response to your major comments. And additional supplemental figures now show the ice mass mixing ratio for all default simulations, as well as the ice crystal radius evolution.

p. 3, line 27: 237 K is not the homogeneous freezing temperature of pure water. The generally accepted value in the literature is 235.15 K. The AMS Glossary of Meteorology states that homogeneous nucleation occurs near 233.15 K.
Thank you for pointing this out. We write "*or a reaches a temperature of 237 K above which no homogeneous nucleation occurs.*"

p. 7, Fig. 2 Captions: How were the values of 2 and 10 fragments per drop chosen? How is the dependence on drop size handled?
Two was chosen as the minimum number of fragments into which a droplet could fragment. Ten was chosen as an upper bound because it represents an order of magnitude increase upon each fragmentation. In what was formerly Equation 2 (now Equation 4), $\aleph_{DS}^{(coll)}$ contains the droplet size

dependence: $\aleph_{DS}^{(coll)} = F_{DS} (2r_R)^4 p_{sh} (T)$. So the fragment number is quartic in droplet size, as in Lawson et al. 2015. This equation was also given in Table S1.

p. 10, Line 17: Lawson et al. 2015 explicitly state that rime splintering is not responsible for the observed secondary ice process. Delete this reference.
Yes, thank you for catching this. Lawson et al. 2015 did emphasize the importance of the liquid phase to secondary ice production, but not to secondary ice production from rime splintering.

p. 11, Lines 4-7: What are the justifications for these assumptions and modifications?
Droplet levitation experiments at the Karlsruhe Institute of Technology are the basis for these modifications to the fragment generation function. In particular, these experiments indicate that the Lawson et al. parameterizations underestimates the fragment number generated for smaller droplets (D ~ 100 um) and overestimates the number for larger droplets (D ~ 1 mm). The sigmoid function addresses both of these concerns. Changing the exponent in the polynomial form addresses a potential overestimation for larger droplets only.

In Table S1, where we give the explicit functional forms of these modified fragment generation functions, we cite "Droplet levitation experiments", but we also point this out in the text now.

p. 12, Lines 1-5: The production of ice in this scenario may be of some interest, but of more interest to cloud physicists is how the ice and water mass budgets evolve. Please show these.

p. 14, Line 5: This is the first mention of CCN. Were CCN used in the model, and if so, how?
To the statement that *"the droplet generation function consists of droplet activation, calculated from a Twomey power-law formulation"*, we have added in Section 2 that "*droplet number is calculated solely from supersaturation rather than a CCN number*" because aerosol is not treated explicitly in our framework.

p. 16, Line 8: "warm cloud base". All cloud bases cited in the paper < 273 K, so there are no warm cloud bases.
Yes, accurate wording here would be "***warmer** cloud base"*, i.e., those parcels that are initiated from relatively warmer subzero temperatures. We have changed this to "*warmer subzero cloud base temperatures*" in a few places.

**Table 2.** Comparison of parcel model results in each section with results from in-situ and laboratory measurements not used to constrain the model formulations.

| | *In-situ measurements* | *Laboratory studies* | *Parcel model simulations* |
|---|---|---|---|
| ***Temporal evolution of*** $N_i$ | **BR** and **DS**: 20 min to form drizzle drops and 12-15 min to glaciation after first ice (Taylor et al., 2016); **DS**: 2-3 min to glaciation after first ice (Lawson et al., 2015); **RS**: $10^2$ enhancements within 10-15 min (Hobbs and Rangno, 1990), 8 $L^{-1}$ over 32 min (Heymsfield and Willis, 2014) | **BR**: 20 min to increase ICNC by a factor of 10 with initial ICNC of 3 $L^{-1}$ (Vardiman, 1978) (his Fig. 7); **DS**: only 50 seconds to fragmentation after equilibration and nucleation time (Johnson and Hallett, 1968); **RS**: Linear increase starting between 10 and 20 min (Hallett and Mossop, 1974) (their Fig. 1) | **BR**: Superexponential increase based on $N_{INP}^{(tot)}$; **DS**: threshold increase based on $p_{fr}$; **DScoll** Exponential increase based on $N_R$; **RS**: Superexponential increase based on $N_R$ |
| ***Limiting INP or thermodynamics*** | **BR**: $T_{top}$ between -10° and -18°C with $N_{ice}$ from 0.1 to 5 $L^{-1}$ (Rangno and Hobbs, 2001); **DS** and **RS**: Taylor et al. (2016) cite the importance of the warm rain process through $T_0$, CDNC, $u_z$, and cell lifetime; **DS**: $N_{INP}$ of $10^{-4}$ to $10^{-2}$ $L^{-1}$ for $\overline{N_i}$ of 572 $L^{-1}$ (Lawson et al., 2015); **RS**: $N_{INP}^{(lim)}$ of 0.01 $L^{-1}$ (Crawford et al., 2012) | **BR**: Strong modulation of ultimate ICNC by initial ICNC (Vardiman, 1978) (his Figure 1); **DS**: $N_{INP}^{(lim)}$ of 1 $m^{-3}$ (Beard, 1992), Favorable temperatures colder than those for RS (Korolev et al., 2004); **RS**: optimal temperatures between -3 and -8°C (Hallett and Mossop, 1974), modest updrafts are most favorable (Mossop, 1985; Heymsfield and Willis, 2014) | **BR**: $N_{INP}^{(lim)}$ from 2 up to 70 $m^{-3}$, possible only at warmer $T_0$ and slower $u_z$; **DS**: no meaningful $N_{INP}^{(lim)}$, favored at colder $T_0$ down to 258 K as $u_z$ slows; **RS**: no meaningful $N_{INP}^{(lim)}$, favored for 268-270 K but this range widens as $u_z$ increases |
| ***Parametric uncertainty*** | **BR**: In-cloud graupel collision rate of 1 $m^{-3}$ $s^{-1}$ (Mizuno and Matsuo, 1992), 10% of ice particles were fragmented (Rangno and Hobbs, 2001); **DS**: $10^{-9}$ fragments per kg liquid (Lawson et al., 2015), 10% of drops frozen by -6°C (Brownscombe and Thorndike, 1968); **RS**: 1.4 $L^{-1}$ $s^{-1}$ (Taylor et al., 2016), 50 crystals $s^{-1}$ (Heymsfield and Willis, 2014) | **BR**: Fragment generation rate $K_0$ of 0.00081 up to 0.01 $L^{-1}$ $s^{-1}$ (Vardiman, 1978); **DS**: Shattering frequencies of 10 to 37% between 50 and 120 $\mu$m (Brownscombe and Thorndike, 1968; Takahashi, 1976); **RS**: 250-700 splinters per mg rime at $u_z$ = 1.5 m $s^{-1}$, 200-400 at 2 m $s^{-1}$ (Hallett and Mossop, 1974) (their Fig. 3), 90-350 (Mossop, 1985) | **BR**: $10^2$-fold enhancement increasing $F_{BR}$ from 40 to 280 at $N_{INP}^{(tot)}$ of 0.17 $L^{-1}$; **DS**: 10-fold enhancement increasing $p_{sh}^{(max)}$ from 1 to 30% independent of $N_{INP}^{(tot)}$; **RS**: 10 minute sooner enhancement increasing $F_{RS}$ from 3 x $10^8$ to 3 x $10^9$ for all $N_{INP}^{(tot)}$ |

---

## Author Comment (AC2) · 19 Aug 2017

**Responses to Reviewer 2 Comments**

The manuscript investigates the role of three secondary ice production mechanisms (rime splintering, frozen droplet shattering, and breakup), more specifically the evolution of the total ice number concentration depending on secondary ice production, the thermodynamic limitations on the secondary processes and the dependence on the chosen parameterization. The authors found that the evolution of the total ice number concentration is determined by the involvement of two phases and the non-linearity of the collision process. However, in case all processes are active none of them is dominant over the others. They also found that only breakup needs a minimum number of ice nuclei, all other processes are more sensitive to the number of cloud condensation nuclei and thermodynamic conditions. The results are summarized in Fig. 8 where they show in which thermodynamic region and also for which ice nuclei concentration and rates secondary ice formation is favorable. The manuscript adds some interesting aspects to the question what bridges the gap between ice nuclei and ice crystal measurements.

We thank the reviewer for the careful reading and thoughtful feedback.

**Major Comments**

The simulations show some interesting aspects of secondary ice production. However, the interpretation aspect could be stronger emphasized. What do the findings have for consequences in terms of modeling of mixed-phase clouds? How are the results connected to field observations? Do the findings agree with observations? Do the findings make sense in the general context / understanding of the microphysics of a mixed-phase cloud? Which further aspects would need to be investigated?

We have added a section before the final summary and outlook called "Comparison with experimental studies". It includes the table at the end of these responses to link field / laboratory measurements with our results in each section. The text in this section also more thoroughly addresses these interpretation questions:

"In Table 2, the parcel model results from the previous three sections are compared to those from field and laboratory measurements for each process. Similar time frames for enhancement, O(30 minutes), and favorable modest updrafts and warmer cloud base temperatures are present in both the observations and simulations. The same importance of  $N_{INP}^{Iim}$  to collisional breakup has been shown in other studies (Vardiman 1978).

An important limitation of the parcel model is the assumption of monodispersity. Large droplets shatter more effectively, and large graupel will have a larger sweep-out kernel for collisional breakup. Without the tails of a hydrometeor size distribution, these larger hydrometeors are omitted, and secondary production is underestimated. An ongoing study will implement similar formulations into a mesoscale meteorological model to understand the effect of this assumption. Ventilation effects, spatial phase separation, and continuous sedimentation are other, more advanced features of a real-world parcel that could also alter these parcel model results [Sullivan et al 2017]. For example, droplet or ice hydrometeor growth will be enhanced by the stronger vapor density gradient generated by their incloud motion. Again, omitting additional hydrometeor growth will underestimate secondary production. If `pockets' of ice phase exist within mixed-phase cloud, then the values of  $N_{INP}^{Iim}$  will be more influential as the collisional breakup contribution will increase relative to rime splintering or droplet shattering. If a continuous formulation of sedimentation were substituted for the threshold one used here, the largest enhancements in Figures 3 and 4 should shift to higher updrafts. Large hydrometeors would be held aloft by these higher updrafts and feed into the secondary production tendencies. The parcel model could be extended in future studies to investigate these effects." In regard to future aspects to consider, we also note at the end of Section 3.1, that "future work should also incorporate a dependence of  $p_{fr}$  on the number of submerged INP, rather than just on time and temperature."

**Specific Comments**

- Page 2, line 23-26: Could you not calculate the number of INP from the nucleation rates? Yes, you're correct. The absolute INP number could be calculated from the adjusted primary nucleation rates and the simulation duration. However, investigating the effect of primary nucleation rate on secondary production is still different than investigating the effect of the absolute INP number on secondary production. The wording about the Connolly et al. 2016a study is adjusted: "Connolly et al. 2006a found that rime splintering production increased with increasing primary nucleation rate but did not give an estimate of a threshold NINP for this rime splintering to initiate."
- Page 3, line 4: How do you derive a nucleation rate from the INP concentration given in DeMott et al. 2010?

Some details of the model construction were omitted because they are covered extensively in another *Journal of Geophysical Research* manuscript (doi:10.1002/2017JD026546). The manuscript recently became available, and we have included more details in the revised manuscript. In particular, here, we note: "*More specifically, the nucleation rate is calculated as the product of updraft velocity, an assumed lapse rate of 6 K km-1, and the temperature derivative of the INP concentration:*  $u_z \Gamma d/dT [a_1 exp [a_2 (T - a_3)]$ ."

- Page 3, line 4: Why a Heaviside function? The Heaviside function is used to "transfer" hydrometeors that have had sufficient time for depositional and riming growth to the next highest size bin. But this time delay Heaviside is applied later to all terms in the generation function. So Equation 1 is corrected by omitting the *H*(*t*).
- Page 3, line 9: How is the connection between *DS* and *DScoll*? From the description (*DScoll* = collision between large droplet and ice crystals?) it sounds like two different processes. Add more explanation to this point.

*DS* refers to the shattering of a droplet as it begins to freeze because of a submerged ice nucleating particle. *DScoll* refers to the shattering of a droplet after collision with an ice crystal. Indeed, these could be considered two different processes that act simultaneously. For now and for comparison's sake, we have only allowed one or the other to be active and label both as "droplet shattering" since that is the mechanism (just induced by different phenomena).

The following clarification is added to the manuscript: "Later, droplet shattering is induced by collision with an ice crystal (denoted DScoll), rather than by internal freezing on a submerged INP. In this case, the tendency includes a product of large droplet and ice crystal numbers."

- Page 3, line 9: Why is it 1% and not 0% outside of the temperature range? This is done to account for uncertainty about whether this optimal temperature range of -3 to -8°C is for the rimer surface temperature or ambient cloud temperature (e.g., Heymsfield and Mossop 1984). We assume that even when the ambient cloud temperature is lower, there may be locally warm enough regions on the non-leading edge of the hydrometeor for rime splintering to occur.
- Page 5, lines 3-4: Explain that more explicitly, example? This section is organized so that this initial idea, *"structure ... can be understood by considering whether the process is collisional and whether it involves ... one or both phases"*, is explained in the

proceeding paragraphs. To make this organization clearer, the following changes are made to the topic sentences:

"When the process involves a product of hydrometeor numbers, as for breakup and rime splintering, the  $N_{ice}$  evolution is non-linear."

"When the process involves a single hydrometeor number, as for this case of droplet shattering, the Nice evolution is linear and does not grow steadily. Instead it exhibits threshold behavior when the temperature becomes cold enough for a non-negligible freezing probability according to Bigg 1953."

"Because breakup and rime splintering involve the ice phase, increasing  $N_{INP}^{tot}$  boosts their rates and yields large enhancements sooner."

"Finally non-linearity and hydrometeor phases involved determine enhancement timing."

• Page 7, Equation 2: What is the physical concept or idea behind this formula or approach? This equation is a reformulation of the ice crystal generation from droplet shattering, i.e., a replacement for the final term in Equation 1. It can then be interpreted like the other terms in Equation 1, where  $K_{DS}$  is a gravitational collection kernel,  $\aleph$  is the generated fragment number, and  $N_R$  and  $N_i$  are large droplet and small ice crystal numbers respectively. It is exactly analogous to the formulation of the rime splintering terms, but with  $N_i$  rather than  $N_g$  or  $N_G$ . To make this connection clearer, we expand Equation 1 as follows:

$$G_{ice} = \frac{dN_i}{dt}\Big|_{NUC} + \frac{dN_i}{dt}\Big|_{BR} + \frac{dN_i}{dt}\Big|_{RS} + \frac{dN_i}{dt}\Big|_{RS}$$
(1)
$$= c_0 H(t) + \eta_{BR} K_{BR} \aleph_{BR} N_g N_G + \eta_{RS} \aleph_{RS} \Big[K_{RS,g} N_g + K_{RS,G} N_G\Big] N_R + \eta_{DS} \aleph_{DS} N_R$$
(2)

(This adjustment makes the collisional droplet shattering tendency Equation 3.) Before the equation for collisional droplet shattering, we add

"the droplet shattering tendency is adjusted to be proportional to both  $N_R$  and  $N_i$ , rather than just  $N_R$  as in the final term of Equation 2:"

- Page 9, line 2: Where do I see that in the figure? "For droplet shattering and rime splintering, there is no  $N_{INP}^{Iim}$  value greater than  $1 m^{-3}$ " because from the lowest value of  $N_{INP}^{tot}$  on the y-axis of Figures 3 and 4 (= 1 m-3), there is a color corresponding to a 100-fold ice crystal number enhancement or higher. Large enhancements are occurring from these processes even at low INP numbers. To the text, we add "there is no  $N_{INP}^{Iim}$ value greater than  $1 m^{-3}$ : the enhancement is largest at the lowest value of  $N_{INP}^{tot}$  in Figure 3 and decreases with higher values of  $N_{INP}^{tot}$ ."
- Page 9, line 9: Again where and how do I see that in the figure? Yes, thank you for pointing out that this result (*"breakup remains the only process with a defined NINPlim"*) is not immediately clear from the figure. We add the following: *"At the lowest values of NINPtot for the droplet shattering and rime splintering panels, enhancements are large and decrease with increasing NINPtot. Only for breakup does NINPtot need to surpass a threshold before a large enhancement occurs."*
- Page 10, line 14-17: What is the reason for these differences?

In fact, we want to mention that previous studies *agree* with our results here. The Connolly et al. study produced a decrease in ice production from freezing rain when primary nucleation was more efficient, and we also see a decrease in production from droplet shattering with additional nucleation. Then the Mossop, Hobbs and Rangno, etc. studies emphasize the importance of initial liquid hydrometeor formation, rather than ice nucleation, to secondary production. We clarify by writing *"Other studies have also emphasized the importance of initial liquid hydrometeor formation, rather than primary nucleation, to ice generation from rime splintering."*

• Section 3.3: Make it clear what process which paragraph is referring to, it starts with Breakup, page 11, line 4 DS...

Thank you for pointing this out. To the start of the first paragraph, we add "First the effect of nucleation rate is investigated on the  $N_{INP}^{lim}$  value for breakup is illustrated in Figure 5. The top panels show results from a default nucleation rate and ones reduced by factors of 10 and 100."

We adjust the start of the third paragraph also: "Then the effect of shattering probability and generated fragment number are investigated for droplet shattering." And the start of the last paragraph: "Finally, we investigate the impact of the fragment number from rime splintering."

- Page 13, line 10: Please also describe what can be seen in panel b. Thank you for pointing this out. The commentary is expanded as follows: "*FRS is the more influential factor in timing. Its impact on Nice evolution is shown in panel b, where a given enhancement is obtained over a shorter period for a higher FRS. As for breakup, the effect of the parameter is much larger when NINPtot is smaller in the yellow traces.*"
- Page 14, line 1: You did not really explain the single versus two-phasedness before? It is an interesting aspect and maybe you could explain that a bit further (here or in the sections before). The notion of single versus two-phasedness is mentioned in the abstract and one of the two key aspects in the analysis of Section 3.1. To make this more clear, we adjust the terminology in the first paragraph of Section 3.1: "The structure in the number evolution can be understood by considering whether the process is collisional and its 'phasedness', i.e., whether it involves hydrometeors in the liquid or ice phase or both."
- Page 14, summary point 1: It could be interesting to illustrate this point in a table or figure. In figure 8, it is not really depicted for each process separately.
  Thank you. We agree and have added a "structure summary" in panel (a) of an additional figure (below).
- Page 14, line 23: What do you mean by emissions? Aerosol emissions? Yes, aerosol emissions. This is added in.
- Page 14: It could be interesting to plot the dominant regions of each process on a 2D-Plot with the vertical velocity and the temperature on the axis.
  Thank you. We agree and have added a "thermodynamic summary" in panel (b) of an additional figure:

• Page 15, line 13: You could add more references here, e.g., Conen et al. 2011 and Steinke et al. 2016.

Thank you. These references have been added in here.

- Figure 1: Panel d is not mentioned / explained in the text. Either remove it or explain it in the text as well.
  Thank you for noting this. To the first paragraph of Section 3.1, the following is added: "These varying values of NINPtot are shown in different colors in panel d of Figure 1."
- Figure 2: Explain in the caption what n = 2, n = 10 means (it is only indirectly explained in the text). In fact in the caption, it states "for the main panel (a), droplet shattering generates 2 fragments per collision, and for the inset, 10 fragments per collision." But the n's are changed to  $\varkappa$ 's to agree with the notation in the text, and ' $\varkappa = 2$ ' and ' $\varkappa = 10$ ' expressions are added in parentheses to the caption explanation.
- Figure 3: In the case of *BRth* and *RSth*: does the same argument yield as in *DSth* for choosing a velocity of 0.5 instead of the smallest value of 0.1?
  Yes, exactly. This is added to the caption: "The lowest updraft of 0.1 m s-1 is not shown because only very small enhancements occur. Droplet shattering is shown at 1 m s-1 for the same reason."
- Figure 3: Why are there no meaningful enhancements by breakup if the updraft is larger? Less collisions?

An explanation for no breakup enhancement at larger updrafts is given on page 9, lines 6-7 (in the original manuscript): "In particular, enhancement from breakup disappears for all  $T_0$  values at a larger  $u_z$  because the parcel is too short-lived for graupel to form again." If the parcel does not exist long enough for the depositional and riming growth of small crystals to graupel, then breakup will simply not occur.

 Figure 4: Why are there no meaningful enhancements by breakup at colder T0? Thank you for pointing this out. Here we have added in an explanation at the end of page 9 (in the original manuscript): "no enhancement from breakup occurs again because graupel does not form. In this case, not only is the parcel too short-lived for graupel formation; diffusional growth is also slowed significantly at such low temperatures."

- Figure 8: It is a nice summary of the outcome of the paper and can be quite a useful Figure. You could strengthen that a bit more. In the current version of the manuscript is not very prominent. We include additional mentions of this figure earlier in the results section.
- Figure 8: What is meant by diminished INP efficiency? The idea of diminished INP efficiency is discussed on page 10, line 10 of the original manuscript, i.e., that the largest ice crystals enhancements are at the lowest  $N_{INP}^{tot}$  values and that as the INP concentration increases the secondary enhancement per INP, a kind of efficiency, drops. This term is highlighted in Section 3.2 to draw attention to it, and it is mentioned again in the Figure 8 caption.
- Figure S1: You could add *BR*, *RS*, and *DS*. Ok, the abbreviations have been added in.
- Figure S1: The process rime-splintering is not clearly depicted (how does the ice multiplication happen?).
  The figure has been adjusted to show a rimed graupel intermediate between the colliding droplet and graupel and the secondarily-produced crystals.

**Small remarks, typos:**

- The INP subscript is not nice to read. Reduce the space between the letters or write it non-italic (which is standard for physical subscripts?).
   Ok, the INP subscript has been un-italicized throughout.
- Page 1, line 20: Year missing at citation Ladino et al. Thank you. '2017' added to the Ladino citation.
- Page 2, line 28: Delete above. Ok.
- Page 4, line 3: Replace freezing by melting. 272 K is the melting temperature. Freezing normally happens at lower temperatures.
  Ok.
- Page 4, line 6: Also FDS and FRS?
  Yes, thank you. FRS was not mentioned and FDS was not listed next to its description. These have been added in.
- Page 4, line 7: The parameters for the functional form are β and γ, if yes add in brackets after "... per shattering droplet"
  Yes, thank you. FDS (β, γ) is listed thereafter now.
- Page 5, line 13: Remove brackets around citation. Ok, thank you.
- Page 5, line 15: Add "explained in" before Section ... Ok.
- Page 5, line 25: What does "its magnitude" refer to? The magnitude of the enhancement. The wording in this topic sentence has been changed.

- Page 5, line 30: Has to be Nicemax instead of NINPmax? Yes, thank you for your careful reading.
- Page 7, line 16: Add brackets around the citation "(Paukert et al. 2017)" instead of "Paukert..." Ok.
- Page 14, line 19: Should or is? Yes, *is* is appropriate. The wording is changed.
- Page 14, line 28: The brackets are strange. These are fixed.
- Page 15, line 2: The term "supercooled liquid fraction" might need a sentence of explanation. The wording is changed to "the liquid portion of a mixed-phase cloud could also..".
- Page 16, line 22: This is also a leading coefficient? Yes, the wording is changed to correspond to that for *F*BR and *F*DS: "Leading coefficient of the fragment number per kilogram of rime..."
- Page 17, line 17: *D* (diameter) is missing in the variable list. Either add it or exchange it here with *r*.
  The diameter variable is exchanged for the radius.
- Table 1: In the Caption there is a run mentioned denoted INP below, which does not exist in the Table?
  Yes, thank you. The *INP* run is not described in the table, only shown in Figure 1d. The wording in the caption is fixed to reflect this.
- Table 1: Reformulate in the Caption (since thermodynamic simulations is ambiguous, *BRth* etc. is also a thermodynamic simulation): "Thermodynamic simulations run with ... are shown solely ..." Ok, thank you.
- For the simulations only shown in the Supplement (*BRDSth...*) no Table with conditions of the simulations exists. However this could be helpful in comparison to Table 1.
  A Table S2 has been added to supplementary information with the details of the BRRS, BRDS, DSRS, BRRSth, BRDSth, and DSRSth simulations, and the caption of Table 1 is modified to note this.
- Table 1: Run *DSpp*: What does the *D* mean in the range of values for *FDS*? *D* indicated the diameter of the freezing droplets. Since the *D* is replaced by *r*, as you noted above, the same is done here. This is corrected throughout Table S1 as well.
- Table S1: *FRS* and *FDS*: What is frag? Fragments? Yes. This abbreviation is removed from the *FRS* and *FDS* expressions to avoid confusion.
- Table S1: psh: What is N?
  N indicates the normal distribution. This is now noted in the caption.
- Table S1: pfr: What is A?
  A is a constant. It is substituted for its specific value, 0.82.

- Legend Figure 5 (a), (b), (c), Fig. 6 (a) and Fig. 7 (a) needs to be bold to be consistent (matters most in case of Fig. 5).
  The letters of these subpanels have been bolded.
- Figure 5 a, b, c: It is a bit unlucky that most points are in the bluish range of the color scale. It is quite difficult to differentiate the different color tones of blue. Independent of what color scale is used, the points will fall in a similar color range and be difficult to differentiate because they correspond to similar values.
- Figure 5 and 6 and 7: Is the color scale in the legend the same for the green traces and the yellow traces (also only color of green traces is shown)? Mention in the Caption or add the colors for the yellow traces also to the legend.
  Yes, the gradient from dark to light in both green and yellow corresponds to the same parameter values. There is not room in the subpanels to include a larger legend, so this is added to the caption.
- Figure 7: The coloring is only similar to Fig. 1 c) for the first color of the green and yellow traces? I found this comment a bit confusing and it did not add necessary information, so maybe delete it. Ok, it has been deleted.
- Figure 8: Are the arrows from one panel to the other needed? What do they symbolize? For now, we have left these in place, as an indication that each panel contains subsequent "criteria" for secondary production to occur.
- Figure 8: What is *s*M? What is *D*V? (left panel) These variables are the supersaturation with respect to water and the water vapor diffusion coefficient in air. Both have been defined in the Notation Appendix.
- Figure S2 Caption: line 1: add on *BRpp* in the end. This figure shows the effect of adjustments for the parameter perturbations with both breakup and droplet shattering, so "*BRpp and DSpp*" are added to the caption.
- Figure S2 Caption: line 2: (b) shows the effect of the minimum... function on... Ok.
- Figure S2 Caption: line 3: ... droplet due to FDS?
  The caption text is adjusted: "Panel (c) shows the effect of the leading coefficient within the droplet shattering fragment generation function, while ..."
- Figure S2 Caption: (d) What is plotted here? Freezing probabilities? Does not fit to the plot. The caption states that panel (d) *"shows various temperature-dependent freezing probability distributions."* All temperature-dependent curves in this Figure are shown for a temperature range of 237 to 273 K since these are the only relevant values for secondary production.
- Figure S3: Replace um with μm. Ok.
- Figure S3 Caption: Add: in dependence of *DR*. Ok.
- Figure S3 Caption: How is the second sentence connected to this figure? Yes, you're right. This is not relevant here; it has been removed.

- Figure S4 Caption: Shift bracket behind "number". Ok.
- Figure S7: Difficult to read legend (a). The coloring here makes the legend quite difficult: either black will not be visible on the blue or white will not be visible on the cyan. We tried to compromise with a gray.

|                          | In-situ measurements                                                   | Laboratory studies                                                   | Parcel model simulations                                                       |
|--------------------------|------------------------------------------------------------------------|----------------------------------------------------------------------|--------------------------------------------------------------------------------|
|                          |                                                                        |                                                                      |                                                                                |
| Temporal evo-            | BR and DS: 20 min to form drizzle drops                                | BR: 20 min to increase ICNC by a fac-                                | BR: Superexponential increase based on                                         |
| lution of N i | and 12-15 min to glaciation after first                                | tor of 10 with initial ICNC of 3 $\rm L^{-1}$                        | $N_{\mathrm{INP}}^{(tot)}$ ; DS: threshold increase based on                   |
|                          | ice (Taylor et al., 2016); DS: 2-3 min to                              | (Vardiman, 1978) (his Fig. 7); DS: only                              | $p_{fr}$ ; DScoll Exponential increase based                                   |
|                          | glaciation after first ice (Lawson et al.,                             | 50 seconds to fragmentation after equili-                            | on $N_R$ ; RS : Superexponential increase                               |
|                          | 2015); RS: 10 2 enhancements within 10-                     | bration and nucleation time (Johnson and                             | based on N R                                                        |
|                          | 15 min (Hobbs and Rangno, 1990), 8                                     | Hallett, 1968); RS: Linear increase start-                           |                                                                                |
|                          | L-1 over 32 min (Heymsfield and Willis,                                | ing between 10 and 20 min (Hallett and                               |                                                                                |
|                          | 2014)                                                                  | Mossop, 1974) (their Fig. 1)                                         |                                                                                |
|                          |                                                                        |                                                                      |                                                                                |
| Limiting INP             | BR: $T_{top}$ between -10° and -18°C with                              | BR: Strong modulation of ultimate ICNC                               | BR: N (lim) INP from 2 up to 70 m -3 , possi- |
| or thermody-             | $N_{\rm lim}$ from 0.1 to 5 $\rm L^{-1}$ (Rangno and                   | by initial ICNC (Vardiman, 1978) (his                                | ble only at warmer $T_0$ and slower $u_x$ ; DS:                                |
| namics                   | Hobbs, 2001); DS and RS: Taylor et al.                                 | Figure 1); DS: $N_{\text{INP}}^{(lim)}$ of 1 m -3 (Beard, | no meaningful $N_{\rm INP}^{(lim)}$ , favored at colder                        |
| nunics                   | (2016) cite the importance of the warm                                 | 1992), Favorable temperatures colder                                 | $T_0$ down to 258 K as $u_{\rm z}$ slows; RS: no                               |
|                          | rain process through $T_0$ , CDNC, $u_x$ , and                         | than those for RS (Korolev et al., 2004);                            | meaningful $N_{\rm INP}^{(lim)}$ , favored for 268-270                         |
|                          | cell lifetime; DS: $N_{\rm INP}$ of $10^{-4}$ to $10^{-2}$             | RS: optimal temperatures between -3 and                              | K but this range widens as $u_x$ increases                                     |
|                          | $L^{-1}$ for $\overline{N_i}$ of 572 $L^{-1}$ (Lawson et al.,          | -8°C (Hallett and Mossop, 1974), mod-                                |                                                                                |
|                          | 2015); RS: $N_{INP}^{(lim)}$ of 0.01 L -1 (Craw-            | est updrafts are most favorable (Mossop,                             |                                                                                |
|                          | ford et al., 2012)                                                     | 1985; Heymsfield and Willis, 2014)                                   |                                                                                |
|                          |                                                                        |                                                                      |                                                                                |
| Parametric               | BR: In-cloud graupel collision rate of 1                               | BR : Fragment generation rate $K_0$ of                        | BR: 102-fold enhancement increasing                                            |
| uncertainty              | $m^{-3}\ s^{-1}$ (Mizuno and Matsuo, 1992),                            | $0.00081$ up to $0.01 L^{-1} s^{-1}$ (Vardiman,                      | $F_{BR}$ from 40 to 280 at $N_{\rm INP}^{(tot)}$ of 0.17                       |
| -                        | 10% of ice particles were fragmented                                   | 1978); DS: Shattering frequencies of 10                              | L-1; DS: 10-fold enhancement increas-                                          |
|                          | (Rangno and Hobbs, 2001); DS: 10-9                                     | to 37% between 50 and 120 $\mu m$ (Brown-                            | ing $p_{sh}^{(max)}$ from 1 to 30% independent                                 |
|                          | fragments per kg liquid (Lawson et al.,                                | scombe and Thorndike, 1968; Takahashi,                               | of N (tot) ; RS: 10 minute sooner enhance-                          |
|                          | 2015), 10% of drops frozen by $-6^\circ C$                             | 1976); RS: 250-700 splinters per mg rime                             | ment increasing $F_{RS}$ from 3 x 10 8 to 3 x                       |
|                          | (Brownscombe and Thorndike, 1968);                                     | at $u_x = 1.5 \text{ m s}^{-1}$ , 200-400 at 2 m s -1     | $10^9$ for all $N_{\rm INP}^{(tot)}$                                           |
|                          | RS : 1.4 L -1 s -1 (Taylor et al., 2016), | (Hallett and Mossop, 1974) (their Fig. 3),                           |                                                                                |
|                          | 50 crystals s -1 (Heymsfield and Willis,                    | 90-350 (Mossop, 1985)                                                |                                                                                |
|                          | 2014)                                                                  |                                                                      |                                                                                |
|                          |                                                                        |                                                                      |                                                                                |

Table 2. Comparison of parcel model results in each section with results from in-situ and laboratory measurements not used to constrain the model formulations.

---

## Author Response (AR1)

**Initiation of secondary ice production in clouds**

[revised manuscript text omitted]

Another hypothesized mechanism is the shattering of droplets with a diameter of 50 to 100s of $\mu$m upon freezing (Mason and Maybank, 1960; Leisner et al., 2014; Lawson et al., 2015)[c1](Scott and Hobbs, 1977; Phillips et al., 2001; Fridlind et al., 2007). At sufficiently cold temperatures, latent heat release leads to the formation of a liquid core-ice shell structure that eventually shatters upon internal pressure build-up. A third mechanism, independent of the liquid phase, is breakup upon mechanical
15  collision of ice hydrometeors. Vardiman (1978) calculated the fragment number generated during [c2]collisional breakup from a change in momentum, and Takahashi et al. (1995) later conducted experiments with a rotating ice sphere in a cloud chamber to estimate the number of ice crystals ejected versus temperature. Yano and Phillips (2011), and more recently Yano et al. (2015), have identified 'explosive regimes' defined by non-dimensional parameters, where [c3]collisional breakup may enhance ICNC by as much as $10^4$.
20  Laboratory and in-situ data of these processes[c4] are difficult to obtain, and their fragment generation functions and temperature dependence remain uncertain (Field et al., 2017). [c5]Given these uncertainties, implementation of secondary ice production parameterizations in large-scale models would be premature. Instead, small-scale [c6], more controllable model[c7]s provide[c8] a [c9]means of estimat[c10]ing variability [c11]in output secondarily-produced ICNC with these parameters, [c12]as well as [c13] the minimum number of INP needed to initiate secondary production. [c14]This latter variable is called $N_{\mathrm{INP}}^{(lim)}$ hereafter.
* * *
[c1] *SCS*: *Text added.*
[c2] *SCS*: *Text added.*
[c3] *SCS*: *Text added.*
[c4] *SCS*:
[c5] *SCS*: *Text added.*
[c6] *SCS*: *Text added.*
[c7] *SCS*:
[c8] *SCS*:
[c9] *SCS*:
[c10] *SCS*:
[c11] *SCS*:
[c12] *SCS*: *Text added.*
[c13] *SCS*:
[c14] *SCS*: *Text added.*

Some previous studies have estimated $N_{\text{INP}}^{(lim)}$ on the basis of in-situ data. For example in a study of ice initiation in cumulus, Beard (1992) found that a nucleated ICNC of 0.001 $\text{L}^{-1}$ could trigger raindrop freezing around -5°C . More recently, Crawford et al. (2012), with Aerosol Properties, PRocesses And InFluenceS (APPRAISE) campaign data, and Huang et al. (2017), with Ice and Precipitation Initiation in Cumulus (ICEPIC) campaign data, identified a primarily nucleated ICNC of 0.01 $\text{L}^{-1}$ as sufficient to initiate rime splintering. [c15]Connolly et al. (2006a) found that rime splintering production increased with increasing primary nucleation rate but did not give an estimate of a threshold $N_{\text{INP}}$ for this rime splintering to initiate. Clark et al. (2005) also adjusted the primary nucleation rate relative to the rime splintering one, but gave no approximate $N_{\text{INP}}^{(lim)}$ values or thermodynamic constraints. These studies have also considered only rime splintering, despite evidence that multiple processes occur simultaneously (Rangno and Hobbs, 2001). We provide more comprehensive estimates of $N_{\text{INP}}^{(lim)}$ here for three secondary production processes [c16] over a range of fragment numbers and thermodynamic conditions.

**2 Parcel model**

To estimate ICNC enhancements and $N_{\text{INP}}^{(lim)}$, we run a parcel model with six hydrometeor classes for small ice crystals and droplets, small and large graupel, and medium and large droplets (Sullivan et al., 2017). [c1]The model predicts the number in these six classes, denoted $N_i$, $N_d$, $N_g$, $N_G$, $N_r$, and $N_R$ respectively. [c2]The hydrometeors in each class are assumed monodisperse, but their sizes are tracked over time as a function of temperature and superaturation. [c3]$N_{\text{ice}}$ is used to denote the summation of the number in the three ice hydrometeor classes. The [c4]bin microphysics consists of primary nucleation and secondary production by [c5]collisional breakup [c6], rime splintering, and frozen droplet shattering. [c7]An ice generation function [c8]is defined to include both primary nucleation and secondary production sources of ice crystals, with units of $\text{m}^{-3}\ \text{s}^{-1}$:

[revised manuscript text omitted]

[c1]*SCS*: *Text added.*

[c2]*SCS*:

[c3]*SCS*:

[c12]*SCS*:

[c13]*SCS*: *Text added.*

[c14]*SCS*: *Text added.*

[c15]*SCS*: *Text added.*

[c16]*SCS*: *Text added.*

**Table 1.** All simulations with parameters adjusted from the default values in Table S1. A control run with no secondary production, i.e., $\eta_{DS} = \eta_{BR} = \eta_{RS} = 0\%$, is denoted INP [c4]in Figure 1. Thermodynamic simulations [c5] run with combinations (BRDSth, BRRSth, and DSRSth) or all (ALLth) of the processes [c6]are shown solely in the Supplement [c7]and detailed in Table S2.

| ***Run BR*** | ***Run DS*** | ***Run RS*** |
|---|---|---|
| | ***(Run DScoll)*** | |
| | Droplet shattering only | Rime splintering |
| [c8]Collisional breakup [c9] | | |
| only | (Collisional droplet shattering only) | only |
| $\eta_{DS} = \eta_{RS} = 0\%$ | $\eta_{BR} = \eta_{RS} = 0\%$ | $\eta_{BR} = \eta_{DS} = 0\%$ |

| ***Run BRth*** | ***Run DSth*** | ***Run RSth*** |
|---|---|---|
| Thermodynamic variations | Thermodynamic variations | Thermodynamic variations |
| for [c10]collisional breakup | for droplet shattering | for rime splintering |
| $u_z = \{ 0.1, 0.5, 1, 1.5, 2, 2.5, 3, 3.5, 4 \text{ m s}^{-1} \}$ | | $T_0 = \{ 256, 258, 260, 262, 264, 268, 270, 272 \text{ K} \}$ |

| ***Run BRpp*** | ***Run DSpp*** | ***Run RSpp*** |
|---|---|---|
| Parameter perturbations | Parameter perturbations | Parameter perturbations |
| for [c11]collisional breakup | for droplet shattering | for rime splintering |
| $F_{BR} = \{0, 90, 140, 200, 280\}$ | $F_{DS} = \{25, 75\}\text{x } 10^{-12}(2\,r_D)^{-4\text{ or }-3}$ | $F_{RS} = \{9, 15, 30, 45, 80\}$ |
| | $(\beta, \gamma) = \{ (-0.016, 500), (-0.015, 400) \}$ | x $10^7$ (kg rime)$^{-1}$ |
| $T_{min} = \{246, 249, 252, ...$ | $p_{sh}^{(max)} = \{1, 5, 10, 20, 30\%\}$ | |
| $255, 258$ K$\}$ | | |

**3.1 Hydrometeor number evolution**

[revised manuscript text omitted]

Although there is no meaningful $N_{\mathrm{INP}}^{(lim)}$ for droplet shattering or rime splintering, $N_{\mathrm{INP}}$ still affects enhancement from these processes. In fact, increasing $N_{\mathrm{INP}}^{(tot)}$ generally decreases enhancement for all $u_z - T_0$ conditions. This can be understood in terms of a sort of **INP efficiency**: the highest ICNC per INP is produced when $N_{\mathrm{INP}}^{(tot)}$ is lowest. Mathematically, increasing $N_{\mathrm{INP}}^{(tot)}$ increases the denominator of the enhancement ratio without a corresponding increase in the numerator. Physically, a higher $N_{\mathrm{INP}}^{(tot)}$ depletes supersaturation more rapidly, as many small ice crystals grow by deposition, or it may keep the parcel warmer with latent heating. Fragment numbers, $\aleph_{DS}$ and $\aleph_{RS}$, also depend on the large droplet radius or rimed mass, which are reduced at lower supersaturation. Previous work corroborates this understanding: Connolly et al. (2006a) found that increasing primary nucleation led to a decrease in the freezing of rain in cloud resolving simulations[c1]. [c2]Other studies have [c3]also emphasized the importance of liquid [c4]hydrometeor formation, rather than primary nucleation, to [c5]ice generation from [c6] rime splintering [c7] (Mossop, 1978, 1985; Hobbs and Rangno, 1985; Heymsfield and Willis, 2014)[c8].

Finally, Figures 3 and 4 show enhancement from a single process, but enhancement from multiple secondary production processes simultaneously can generally be understood as the linear combination of that from these single processes (Figures S8, S9, or S10). For example, the pattern of enhancement from ALLth in Figure S8 looks like the addition of the patterns from RSth, DSth, and BRth in Figure 3.
* * *
[c5]*SCS*: *Text added.*
[c6]*SCS*: *Text added.*
[c7]*SCS*: *Text added.*
[c1]*SCS*: ,
[c2]*SCS*: while many
[c3]*SCS*: shown
[c4]*SCS*: phase properties
[c5]*SCS*: *Text added.*
[c6]*SCS*: the
[c7]*SCS*: tendency
[c8]*SCS*: (Lawson et al., 2015)

[revised manuscript text omitted]

---

## Editor Decision (ED1)

MAJOR COMMENTS (copied from initial review for clarity)

I will start this review by confessing that I am an observationalist, not a modeler. I bring an obvious bias into this review, which is that I couch my evaluation of this work in terms of data collected in clouds, not numerical simulations of clouds. My main concern with this manuscript is that I cannot determine how this parcel model relates to an updraft in a cloud. Presumably, a parcel model is intended to represent the evolution of an undiluted parcel of cloud as it rises in the atmosphere. However, it is not clear what the prognostic microphysical variables are in the model. Presumably the model predicts $N_{ice}$ for each of the categories because this is shown in the "ice generation function" equation, but what about mass? It's not even clearly stated if it's a bulk or bin microphysics scheme. It's also not clear what the "ice generation function" itself is, and what the units of $G_{ice}$ are. I'm assuming this is $dN_{ice}/dt$ from all microphysical processes, but it's not clear. The Sullivan (2017 – JGR) reference is in review and of no help. Even if the Sullivan JGR paper becomes available, at a minimum the manuscript should state what the model predicted variables are, and how they are being solved in the model numerically (e.g., what kind of time stepping method, the time step, etc.). It would also help if the manuscript gave the evolution equations for the model predicted variables.

The manuscript shows no drop or ice particle size distributions and no liquid water or ice water contents as a function of temperature. Also, the observations that I am most familiar with suggest that clouds with cloud-base temperatures colder or equal to 0 C, which are all of the cases examined here, do not produce cloud drops large enough to support drop shattering, and generally not even large enough to support rime splintering. Large drops (drizzle and rain drops) are what the literature (e.g., Koenig 1963, 1965; Hobbs and Rangno 1990, Rangno 2008, Lawson et al. 2015) associates with drop shattering and rapid glaciation. The data suggest that the formation of millimeter-diameter supercooled drops requires cloud base temperatures warmer than approximately +18 C (291 K) and broad (> 50 µm diameter) cloud base drop distributions. Albeit, the requisite relationship between CCN and cloud base temperature is yet to be accurately quantified. Also, the coalescence process is key to the formation of supercooled large drops. Nowhere in the manuscript can I find how coalescence is handled in the model (except that $K_x$ is a gravitational collection kernel in Eq. 1). One aspect of the simulations that does appear to be consistent with the observations is that rime-splintering takes place only in clouds with very weak updrafts (e.g., Heymsfield and Willis 2014). However, it is not clear in the manuscript exactly why this takes place in the simulations. Before I can recommend publication, the manuscript needs to provide an explicit description of the model, and the evolution of the parcel in terms of microphysical parameters (liquid and water size distributions, LWC, IWC as a function of temperature). I understand that this may be a bit artificial given the six categories of particles, but an attempt must be made, and the results should be compared with observations.

MAJOR COMMENTS (from review of revised manuscript)

The authors have adequately explained several of the assumptions governing the simulation that were not previously elucidated. This helps the reader to understand the mechanics of the model. They have also addressed several, but not all of the concerns expressed in my previous review. However, one very basic and important point is not addressed in the revision. The paper does not adequately represent observations, and it does not adequately

explain why the simulations differ from observations.   In particular, I point to a recent paper (Lawson et al. 2017 September JAS) showing that the rapid formation of ice in convective clouds is a strong function of cloud base temperature, and that secondary ice via drop shattering does not occur at cloud base temperatures colder than about 273 K.  The simulations suggest that (p. 18 line 8) drop shattering (DS) occurs at "simulated cloud base" temperatures of 260 K and warmer.  Data in Fig. 8 of Lawson et al. (2017) show that a cloud with a base temperature of about 259 K does not produce drops larger than 40 microns, and there is no indication of secondary ice production.  Indeed, supercooled liquid water is measured at 237.7 K.  Also, in their reply and the manuscript (Section 3.1) the authors state that

*"We show the $N_{ice}$ evolution from a `warm-base-convective' sensitivity run in Figure S7. Here the same threshold behavior occurs once the parcel reaches cold enough temperatures for droplet freezing, but there is no $N_{ice}$ decrease beforehand because ice nucleation begins later, and no graupel has begun to fall out."*

In contrast, the observations suggest that the probability of secondary ice production increases proportional to the fourth power of drop radius.  Starting at a warmer cloud base means there is more cloud depth for the coalescence process to occur, which results in the formation of larger drops and a higher probability of drop shattering and ice production.  As far as I can tell this is not represented at all in the simulations.  Also, the observations suggest that the DS process does not depend on the formation of graupel.

Perhaps due to the artificial nature of the model, which assumes six hydrometeor categories that are each monodisperse, the simulations cannot hope to reliably represent the observations.  If that is the case, then the authors need to compare the simulation results with recent observations and explain why the model differs. At the very least, the paper needs to adequately explain how the assumptions in the model impact the results.  Or conversely, since we also know that observations are not perfect due to instrumentation uncertainties and under sampling (i.e., in situ instruments only measure a tiny fraction of the cloud and do not represent a true Lagrangian view of the updraft), the manuscript should explain why the observations are not representative of reality.

Since this manuscript will eventually be published, my desire is that the authors take my criticisms in the manner they are intended, i.e., to improve the paper by including possible counterpoints to their arguments, which are largely focused on the model results and not always on how well they represent real clouds.  We don't know how the actual process(es) of secondary ice production will be revealed in the next few years, or decades, when both measurements and models improve, so reporting both the model results and other possibilities provides a more complete picture.

Several of my previous specific comments were addressed, but some were not.  See the following annotation.

p. 1 Line 4: "Break Up" is not a good term for ice-ice collisions, because drops also break up. I suggest that you find a more descriptive term that applies only to ice. If the term "break up" has to be defined as ice-ice collisions here and everywhere else in the manuscript.

"*Breakup*" was used because preexisting work on this process generally employs this term, e.g., Yano and Phillips *JAS* 2001, Phillips et al. *JAS* 2017, Field et al. *Meteor. Mono.* 2017. But we understand that this terminology may cause confusion with droplet breakup. We have gone through and changed all instances of "*breakup*" to "*collisional breakup*".

Drops collide and can breakup.  Why not call this ice-ice collisional breakup, or at least define it as ice-ice collisions in the beginning of the paper.

p. 2, Line 7: Add references; there are several.
We have added Scott and Hobbs 1977, Phillips et al. 2001, and Fridlind et al. 2007 to the citations for frozen droplet shattering.

In my opinion you have chosen poorly. Scott and Hobbs and Fridland  et al. are mainly theoretical/modeling studies.   There are better papers that deal directly with laboratory experiments and field observations of drop shattering.  The first in situ photograph of a fractured drop, at least to my knowledge, is shown in Cannon et al. (1974).  The photograph was collected from a film camera mounted on a sailplane spiraling in the updraft of a cumulus cloud.  Korolev et al. (2004) showed the first CPI images of fractured drops in both laboratory experiments and from in situ measurements. Rangno (2008) gives a nice summary and shows images of fragmented drops, pointing out that H-M is not active in these convective clouds with secondary ice production.  Wildeman et al. (2017) shows excellent high-speed video of millimeter drops fracturing (photos in his paper; videos on his website). These references are found at the end of this review.

p. 2, Lines 16-17: This is contradictory. In the previous sentence, you reference Field as reporting many uncertainties in the physics of secondary ice production, and then go on to state that small-scale models provide a good tool to estimate variability in secondary-produced ice. The model is only as good as the physics it contains. With the acknowledged vast degree of uncertainties, how can one have any confidence in the model results? If the model results are to be useful, then the physical uncertainties have to be emphasized. Also, sensitivity tests should be run to show how the physical uncertainties impact the results. At a minimum, a disclaimer of this sort needs to be inserted at this point in the manuscript.
We do not believe that these statements are contradictory. Investigating how a given output varies with uncertain parameters is an important application of models. And particularly for small-scale, more controllable models, output variation with adjustable parameters can be well-understood. This kind of work allows experimentalists to focus on measuring the most influential parameters and provides a test-bed for parameterizations prior to implementation in large-scale models. This utility of small-scale models is summarized in the IPCC Assessment Report 5: "high-resolution models enhance our understanding of cloud processes [as] an important tool in testing and improving parameterizations of cloud-controlling processes."
As you note, sensitivity tests should be run with the small-scale model to understand the process and parametric uncertainties. Sections 3.1.1 and 3.3 contain these tests. We run simulations for different formulations of the physics of frozen droplet shattering. And then we investigate the sensitivity to adjustable parameters in the fragment generation functions (particularly $F_{BR}$, $T_{min}$, sigmoid versus polynomial forms for droplet shattering, and $p_{sh(max)}$).
We clarify the utility of small-scale models in this paragraph: "*Laboratory and in-situ data of these processes are difficult to obtain, and their fragment generation functions and temperature dependence remain uncertain [Field et al. 2017]. Given these uncertainties, implementation of secondary ice production parameterization in large-scale models would be premature. Instead,*

*small-scale, more controllable models provide a means of estimating variability in output secondarily-produced ICNC with these parameters, as well as the minimum number of INP needed to initiate secondary production."*

This verbiage still does not justify the use of small-scale models if the physics do not adequately represent reality. This is like saying we can see the specimen better with a high-power microscope, but in actuality the specimen is not within the field of view. I suppose this is the basic rift between observationalists and modelers, who call their results data while observationalists call the results output. Since this is a modeling paper, I guess the modelers get to voice their opinion.

p. 3, Eqn (1) and discussion: Eqn (1) is far too arcane to understand what is going on in the model. The reference to Sullivan et al. (2017) is of no use since it is under review. There are several unanswered questions. What are the units of $G_{ice}$? What is the role of coalescence and how is it handled? What is the cloud base drop distribution? Are CCN included? If so, how? Why don't small ice and small drops appear in Eqn (1)? Also, the number of secondary ice particles produced is only one issue. The mass of ice is of equal if not more importance. If large (millimeter-diameter) supercooled drops are rapidly freezing, as seen in the observations, then the *conversion* of water to ice (and eventually back to water in the form of rain), is more significant than the number of ice particles. Show the results also in terms of water and ice mass.

As described above, we have worked to make the model description more clear without restating what has already been published in the model development manuscript. In particular, we have more clearly stated the purpose and the units of the ice generation function and expanded its mathematical explanation with two additional equations. Small ice and droplets do not appear in Equation 1 because they play no role in any of the processes that are a source of small ice crystals.

Then we have emphasized that there are no size distributions involved; the monodisperse radius or axis of each hydrometeor class is evolved in time. The model contains no explicit aerosol. We add this statement and an in-line equation for primary nucleation before the statement that "*the droplet generation function consists simply of droplet activation, calculated from a Twomey power-law formulation.*" So droplet number is calculated from supersaturation rather than a CCN number. Then we have added more detail for the coalescence formulation to Section 2, as discussed in the response to your major comments. And additional supplemental figures now show the ice mass mixing ratio for all default simulations, as well as the ice crystal radius evolution.

The model description is now clearer

p. 3, line 27: 237 K is not the homogeneous freezing temperature of pure water. The generally accepted value in the literature is 235.15 K. The AMS Glossary of Meteorology states that homogeneous nucleation occurs near 233.15 K.

Thank you for pointing this out. We write "*or a reaches a temperature of 237 K above which no homogeneous nucleation occurs.*"

p. 7, Fig. 2 Captions: How were the values of 2 and 10 fragments per drop chosen? How is the dependence on drop size handled?

Two was chosen as the minimum number of fragments into which a droplet could fragment. Ten was chosen as an upper bound because it represents an order of magnitude increase upon each fragmentation. In what was formerly Equation 2 (now Equation 4), $\aleph_{DS(coll)}$ contains the droplet size dependence: $\aleph_{DS(coll)} = F_{DS} (2r_R)_4 p_{sh} (T)$. So the fragment number is quartic in droplet size, as in Lawson et al. 2015. This equation was also given in Table S1.

p. 10, Line 17: Lawson et al. 2015 explicitly state that rime splintering is not responsible for the observed secondary ice process. Delete this reference.

*Yes, thank you for catching this. Lawson et al. 2015 did emphasize the importance of the liquid phase to secondary ice production, but not to secondary ice production from rime splintering.*

p. 11, Lines 4-7: What are the justifications for these assumptions and modifications?

*Droplet levitation experiments at the Karlsruhe Institute of Technology are the basis for these modifications to the fragment generation function. In particular, these experiments indicate that the Lawson et al. parameterizations underestimates the fragment number generated for smaller droplets (D ~ 100 um) and overestimates the number for larger droplets (D ~ 1 mm). The sigmoid function addresses both of these concerns. Changing the exponent in the polynomial form addresses a potential overestimation for larger droplets only.*

*In Table S1, where we give the explicit functional forms of these modified fragment generation functions, we cite "Droplet levitation experiments", but we also point this out in the text now.*

To my knowledge there has not been a quantitative measurement of the number of fragments produced per shattering event in the levitation experiments. I suggest that you present both the estimate you report from the lab experiments and the estimate from Lawson et al. (2015).

p. 12, Lines 1-5: The production of ice in this scenario may be of some interest, but of more interest to cloud physicists is how the ice and water mass budgets evolve. Please show these.

p. 14, Line 5: This is the first mention of CCN. Were CCN used in the model, and if so, how?

*To the statement that "the droplet generation function consists of droplet activation, calculated from a Twomey power-law formulation", we have added in Section 2 that "droplet number is calculated solely from supersaturation rather than a CCN number" because aerosol is not treated explicitly in our framework.*

p. 16, Line 8: "warm cloud base". All cloud bases cited in the paper < 273 K, so there are no warm cloud bases.

*Yes, accurate wording here would be "**warm** cloud base", i.e., those parcels that are initiated from relatively warmer subzero temperatures. We have changed this to "warmer subzero cloud base temperatures" in a few places.*

**References**

Cannon, T. D., J. E. Dye, and V. Toutenhoofd, 1974: The mechanism of precipitation formation in Northeastern Colorado cumulus II. Sailplane measurements. *J. Atmos. Sci.*, **31**, 2148–2151.

Korolev, A. V., M. P. Bailey, J. Hallett, G. A. Isaac, 2004: Laboratory and In Situ Observation of Deposition Growth of Frozen Drops. *J. Appl. Meteor.*, **43**, 612–622.

Rangno, A. L., 2008: Fragmentation of freezing drops in shallow Maritime frontal clouds. *J. Atmos. Sci.*, **65**, 1455 - 1466.

Wildeman, S., S. Sebastian Sterl, C. Sun, and D. Lohse, 2017: Fast dynamics of water droplets freezing from the outside in. *Phys. Rev. Lett*. **118**, 08410.

---

## Author Response (AR2)

**Reviewer 1 Comments**

The authors have adequately explained several of the assumptions governing the simulations that were not previously elucidated. This helps the reader to understand the mechanics of the model. They have also addressed several, but not all of the concerns expressed in my previous review. However, one very basic and important point is not addressed in the revision. The paper does not adequately represent observations, and it does not adequately explain why the simulations differ from observations. In particular, I point to a recent paper (Lawson et al. September *JAS*) showing that the rapid formation of ice in convective clouds is s a strong function of cloud base temperature, and that secondary ice via drop shattering does not occur at cloud base temperatures colder than about 273 K. The simulations suggest that (p. 18 line 8) drop shattering (DS) occurs at "simulated cloud base" temperatures of 260 K and warmer. Data in Fig. 8 of Lawson et al. (2017) show that a cloud with a base temperature of about 259 K does not produce drops larger than 40 microns, and there is no indication of secondary ice production. Indeed, supercooled liquid water is measured at 237.7 K. Also, in their reply and the manuscript (Section 3.1) the authors state that "*we show the $N_{ice}$ evolution from a 'warm-base convective' sensitivity run in Figure S7. Here the same threshold behavior occurs once the parcel reaches cold enough temperature for droplet freezing, but there is no $N_{ice}$ decrease beforehand because ice nucleation begins later, and no graupel has begun to fall out.*"

In contrast, the observations suggest that the probability of secondary ice production increases proportional to the fourth power of drop radius. Starting at a warmer cloud base means there is more cloud depth for the coalescence process to occur, which results in the formation of larger drops and a higher probability of dorp shattering and ice production. As far as I can tell, this is not represented at all in the simulations. Also, the observations suggest that the DS process does not depend on the formation of graupel.

Perhaps due to the artificial nature of the model, which assumes six hydrometeor categories that are each monodisperse, the simulations cannot hope to reliably represent the observations. If that is the case, then the authors need to compare the simulation results with recent observations and explain why the model differs. At the very least, the paper needs to adequately explain how the assumptions in the model impact the results. Or conversely, since we also know that observations are not perfect due to instrumentation uncertainties and undersampling (i.e. in-situ instruments only measure a tiny fraction of the cloud and do not represent a true Lagrangian view of the updraft), the manuscript should explain why the observations are not representative of reality.

Since this manuscript will eventually be published, my desire is that the authors take my criticisms in the manner they are intended, i.e., to improve the paper by including possible counterpoints to their arguments, which are largely focused on the model results and not always on how well they represent real clouds. We don't know how the actual process(es) of secondary ice production will be revealed in the next few years, or decades, when both measurements and models improve, so reporting both the model results and other possibilities provides a more complete picture.

Thank you for your rereading of and feedback on our work. We had included Table 2 to compare observations with our simulations. We have moved the contents of this table into the text of an *Observational comparison* section (§4) in the hope that this format is more accessible.

Thank you also for pointing us to the newly published Lawson et al. article and for your encouragement to revisit the droplet shattering formulation. The original model described in the Sullivan et al. *JGR* manuscript did not include this process, and its representation was the least

refined of the three. So we have reworked it to yield more realistic behavior. Previous simulations were insensitive to initial temperature first because of the extreme temperature dependence of the Bigg 1953 freezing parameterization. To illustrate, here are the output probabilities for droplets of 100 micron and 1 mm diameter:

[Figure]

Independent of the parcel's initial temperature (or droplet size), freezing probability would be quite small until a temperature of at least -20°C. Then because the model only considered coalescence of small and medium droplets, the large drop size increased due only to condensational growth. So the model did not represent the continued coalescence that Lawson et al. 2017 pinpoint as crucial to initiate secondary production.

A framework with three, monodisperse liquid hydrometeor classifications can never fully represent this coalescence, and so in this sense, the model is indeed artificial. However, observed trends can still be qualitatively reproduced by adjusting the droplet freezing probability and large drop size evolution:

(1) We replace the Bigg freezing probability with a formulation based upon work by Paukert et al. 2017 (doi: 10.1002/2016MS000841). We assume that 10% of the ice-nucleating particles, as predicted by the DeMott et al. 2010 parameterization, do not freeze until small droplet collisions form a rain drop; the primary nucleation rates are correspondingly decreased. Then if about 100 such collisions form a raindrop (Paukert et al. 2017 show a two order-of-magnitude difference between the rain drop and collected particle number concentration in their Figure 5b.), the freezing fraction of the rain drop population is given by $10\,N_{INP}/N_d$.

(2) Coalescence of large drops with one another is incorporated. 5% of the large drop population undergoes coalescence per minute. This reduces the large drop number, and the liquid mass is redistributed over the remaining drops. Another important factor is the addition of a size threshold for the shattering probability: it remains zero unless the droplet diameter is greater than 100 um.

These adjustments introduce two new parameters: the number of INP remaining in a raindrop and the percentage of the large drop population undergoing collision-coalescence. With these updates, we have redone the droplet shattering simulations in all three sections and simulate behavior more in-line with observations. Enhancement now only occurs if the parcel is initiated above the freezing level and if the updrafts are somewhat higher, as can be seen in Figure 1b, Figure 2, and Figures 3c and 4c. We also have updated the analysis throughout, noting the importance of some representation of the warm rain process to have realistic $T_0$ dependence (for example, around lines 1 to 10 page 8 and lines 5 to 10 on page 12 among others). Figure 8 and 9 schematics have also been updated to reflect the need for warmer $T_0$.

Several of my previous specific comments were addressed, but some were not. See the following annotation.

p.1, Line 4: Drops collide and can breakup. Why not call this ice-ice collisional breakup, or at least define it as ice-ice collisions in the beginning of the paper.

We understand that more precise terminology is best and have gone through and changed all instances of "*collisional breakup*" to "*ice-ice collisional breakup*". When the process is first introduced on page 2, line 11, we state that it involves "breakup upon mechanical collision of ice hydrometeors."

p. 2, Line 7: In my opinion, you have chosen [your citations] poorly. Scott and Hobbs and Fridlind et al. are mainly theoretical / modeling studies. There are better papers that deal directly with laboratory experiments and field observations of drop shattering. The first in-situ photograph of a fractured drop, at least to my knowledge, is shown in Cannon et al. (1974). The photograph was collected from a film camera mounted on a sailplane spiraling in the updraft of a cumulus cloud. Korolev et al. (2004) showed the first CPI images of fractured drops in both laboratory experiments and from in-situ measurements. Rangno (2008) gives a nice summary and shows images of fragmented drops, pointing out that H-M is not active in these convective clouds with secondary ice production. Wildeman et al. (2017) shows excellent high-speed video of millimeter drops fracturing (photos in his paper; videos on his website). These references are found at the end of this review.

Given that the work is theoretical, we retain the Fridlind et al. citation, as an example of previous modeling work on droplet shattering. We remove the Scott and Hobbs one, as well as the Phillips et al. 2001 one, which deals primarily with rime splintering. Then we add your suggestions; thank you for these, particularly the Wildeman et al. 2017 one.

p. 2, Lines 16-17: This verbiage still does not justify the use of small-scale models if the physics do not adequately represent reality. This is like saying we can see the specimen better with a high-power microscope, but in actuality the specimen is not within the field of view. I suppose this is the basic rift between observationalists and modelers, who call their results data while observationalists call the results output. Since this is a modeling paper, I guess the modelers get to voice their opinion.

One has to begin somewhere in modeling the physics of a system. Certainly the model has many assumptions, but then there is measurement error in in-situ data or simplification in laboratory experiment set-up, as you acknowledge above. Better physical understanding is an iterative process between measurements of increasing accuracy and models of increasing insight in our opinion. And we would hope that, rather than feeling at odds, the observational and modeling communities cooperate in this iterative process. We leave the statement of the study premise as is.

p. 3, Eqn (1) and discussion: The model description is now clearer.

Thank you. We have also moved Table S1 to the main manuscript, so that details of the model formulations are more readily available.

p. 11, Lines 4-7: To my knowledge there has not been a quantitative measurement of the number of fragments produced per shattering event in the levitation experiments. I suggest that you present both the estimate you report from the lab experiments and the estimate from Lawson et al. (2015).

The fragment number formed per frozen droplet is shown from the Lawson et al. parameterization with various leading coefficients in Figure S2c and those based upon droplet levitation experiments are shown for the two sets of parameters used in Figure S3. More rigorous methods of fragment counting are being developed for the droplet levitation experiments, but the sigmoidal functions are based upon the qualitative observation that the Lawson et al. parameterization underestimates fragment number from small droplets (D ~ 100 um).